JCB Journal of Cell Biology

# The GARP complex prevents sterol accumulation at the trans-Golgi network during dendrite remodeling

Caitlin E. O'Brien[1,2], Susan H. Younger[1,2], Lily Yeh Jan[1,2,3], and Yuh Nung Jan[1,2,3]

**Membrane trafficking is essential for sculpting neuronal morphology. The GARP and EARP complexes are conserved tethers that regulate vesicle trafficking in the secretory and endolysosomal pathways, respectively. Both complexes contain the Vps51, Vps52, and Vps53 proteins, and a complex-specific protein: Vps54 in GARP and Vps50 in EARP. In *Drosophila*, we find that both complexes are required for dendrite morphogenesis during developmental remodeling of multidendritic class IV da (c4da) neurons. Having found that sterol accumulates at the trans-Golgi network (TGN) in *Vps54^{KO/KO}* neurons, we investigated genes that regulate sterols and related lipids at the TGN. Overexpression of oxysterol binding protein (Osbp) or knockdown of the PI4K *four wheel drive (fwd)* exacerbates the *Vps54^{KO/KO}* phenotype, whereas eliminating one allele of *Osbp* rescues it, suggesting that excess sterol accumulation at the TGN is, in part, responsible for inhibiting dendrite regrowth. These findings distinguish the GARP and EARP complexes in neurodevelopment and implicate vesicle trafficking and lipid transfer pathways in dendrite morphogenesis.**

## Introduction

Proper wiring of the nervous system depends on the development and maintenance of complex-polarized neuronal morphologies. Neurons initially establish their architectures during embryogenesis, which is followed by an extended postembryonic period during which excess branches are pruned and remodeled into their mature states (Stiles and Jernigan, 2010). Membrane trafficking pathways are essential to establishing and sculpting neuronal morphology during development (Winkle and Gupton, 2016). Secretory vesicles are essential sources of membrane for neurite outgrowth (Vega and Hsu, 2001), and dendrite development is particularly sensitive to inhibition of secretory trafficking (Ye et al., 2007). Endocytosis and recycling pathways are also required for growth factor-mediated branching of dendrites (Lazo et al., 2013) and axons (Ascano et al., 2009). Both the secretory (Wang et al., 2017; Wang et al., 2018) and endocytic (Kanamori et al., 2015; Krämer et al., 2019) pathways also play important roles during developmental pruning and remodeling of dendrites. Numerous mutations in regulators of membrane trafficking are associated with neurodevelopmental disorders (Ivanova et al., 2017; Marin-Valencia et al., 2017; Ouyang et al., 2013; Passemard et al., 2017), highlighting the importance of these pathways in proper nervous system development.

The closely related Golgi-Associated Retrograde Protein (GARP) and Endosome-Associated Recycling Protein (EARP) complexes are conserved membrane tethers that function in the secretory and endolysosomal pathways. They share the common subunits Vps51, Vps52, and Vps53 (Vps for vacuolar protein sorting; Fig. 1 A). This core interacts with either Vps54 to form the GARP complex (Conibear and Stevens, 2000; Siniossoglou and Pelham, 2002; Reggiori et al., 2003) or Vps50 to form the EARP complex (Gillingham et al., 2014; Schindler et al., 2015). The GARP complex primarily localizes to the trans-Golgi network (TGN) where it tethers endosomes and facilitates SNARE complex formation for the retrograde delivery of cargo to the Golgi (Pérez-Victoria and Bonifacino, 2009). The GARP complex is required for proper sorting of various cargos, including the lysosomal hydrolases (Pérez-Victoria et al., 2008), and for secretion of GPI-linked proteins (Hirata et al., 2015). The more recently described EARP complex associates with early endosomes and facilitates Rab4-dependent cargo recycling (Schindler et al., 2015; Gillingham et al., 2014), as well as Rab2-dependent sorting into dense-core vesicles (Topalidou et al., 2016).

Several neurodevelopmental disorders are associated with mutations in GARP and EARP subunits. Mutations in the core components *Vps51* (Gershlick et al., 2018) and *Vps53* (Feinstein et al., 2014; Hady-Cohen et al., 2018) have been identified in patients who suffer from profound developmental delays and progressive postnatal microcephaly. Mutations in *Vps50* have been linked to neural tube defects (Shi et al., 2019). These studies underscore the importance of the GARP and EARP complexes in neurons, prompting our study to examine their function during neuronal development. The dendritic arborization (da) sensory neurons in *Drosophila melanogaster* are a well

[1]Howard Hughes Medical Institute, University of California at San Francisco, San Francisco, CA; [2]Department of Physiology, University of California at San Francisco, San Francisco, CA; [3]Department of Biochemistry and Biophysics, University of California at San Francisco, San Francisco, CA.

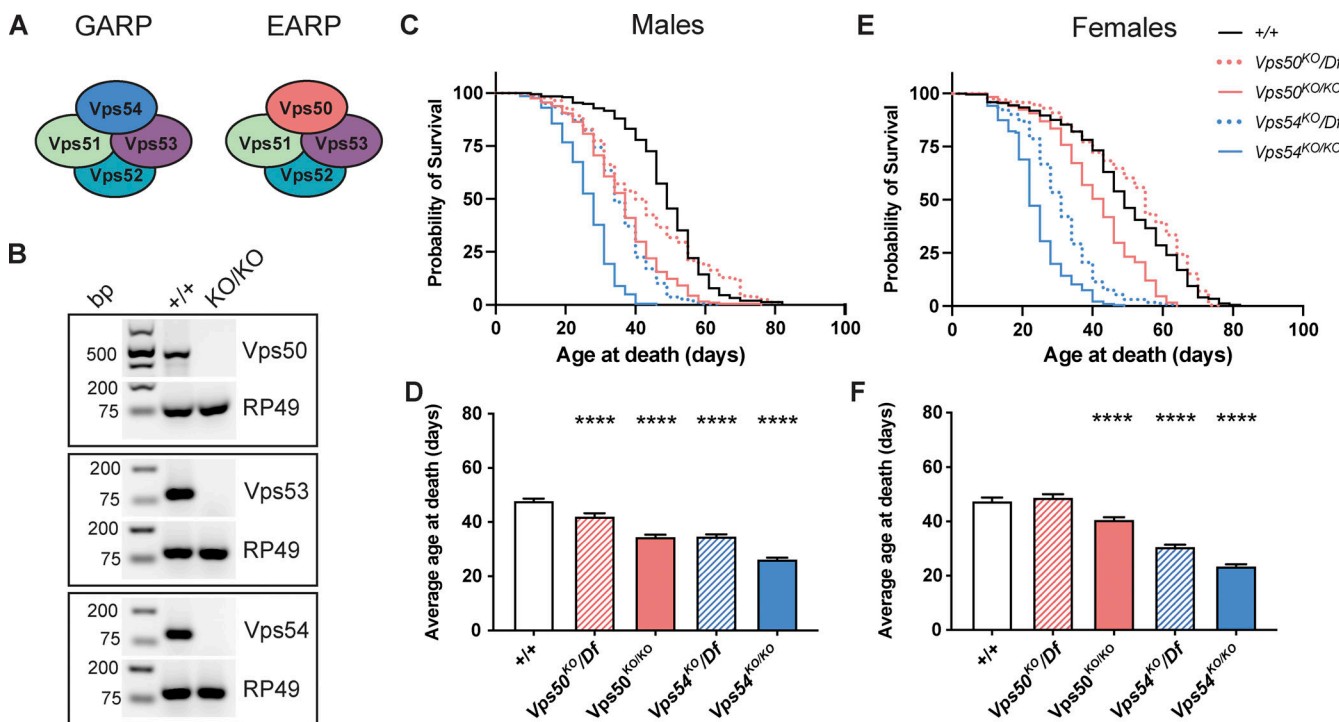

Figure 1.   **Reduced lifespan of GARP knockout flies. (A)** Cartoon depicting the GARP and EARP complexes. **(B)** RT-PCR from control (+/+) and KOs. RNA was extracted from whole larvae, reverse transcribed, and equal amounts of cDNA were used for RT-PCR. RP49 is used as a reference gene. Bp = base pairs. **(C and D)** Survival curves and average age at death ± SEM for male flies of the indicated genotypes. N > 200 flies/genotype. Survival curves were analyzed by Log-Rank Mantel-Cox test with Bonferroni multiple comparisons correction. ****P < 0.0001 for all genotypes compared to control except $Vps50^{KO}/Df$ (not significant—n.s.). Average age at death compared to +/+ was analyzed by one-way ANOVA with Dunnett's post-test. ****P < 0.0001. **(E and F)** Survival curves and average age at death ± SEM for female flies of the indicated genotypes. N > 188 flies/genotype. Survival curves were analyzed as in C. ****P < 0.0001 for all genotypes compared to control except $Vps50^{KO}/Df$ (n.s.). Average age at death analyzed as in D. ****P < 0.0001. Source data are available for this figure: SourceData F1.

characterized model to study dendrite morphogenesis (Grueber et al., 2002). The c4da neurons establish complex larval dendritic arbors, which then undergo developmental pruning and regrowth to their mature adult forms during pupation (Kuo et al., 2005; Williams and Truman, 2005; Shimono et al., 2009), making them amenable to study various aspects of neurodevelopment.

Cholesterol is an important component of cellular membranes, regulating membrane fluidity and protein sorting (Ikonen, 2008; Lippincott-Schwartz and Phair, 2010). While most cells can either synthesize endogenous sterols or obtain them from dietary sources, *D. melanogaster* is a sterol auxotroph—they lack the ability to synthesize sterols—and must obtain sterols entirely from dietary sources (Clayton, 1964). *Drosophila* is therefore an excellent model to study the uptake and transport of sterol between organelles. Dietary sterols packaged with low-density lipoproteins (LDL) are endocytosed after binding the LDL receptor. The Niemann Pick proteins NPC1 and 2 then coordinate the non-vesicular transfer of sterols from the endolysosomal lumen to the ER through inter-organelle contact sites (Höglinger et al., 2019; Infante et al., 2008). Both yeast and mammalian cells lacking the core GARP/EARP components accumulate sterol in lysosomes due to missorting of NPC2 (Fröhlich et al., 2015; Wei et al., 2017). Once in the secretory pathway, sterols are tightly regulated as the flow of cargo through the Golgi is particularly sensitive to sterol levels. Sterol depletion

inhibits secretory vesicle budding from the TGN (Wang et al., 2000), while sterol overload strongly inhibits transport of the model secretory cargo VSV-G (Stüven et al., 2003). During neuronal development, either depletion or overload of sterols can decrease dendrite and axonal branching (Ko et al., 2005; Fan et al., 2002).

In this study, we generated CRISPR knockout flies for shared and complex-specific genes of the GARP and EARP complexes and demonstrate a role for both complexes in the development of adult da neuron arbors. Sterol accumulates in neurons lacking the GARP ($Vps54^{KO/KO}$), but not EARP ($Vps50^{KO/KO}$), complex during regrowth after developmental pruning. Unexpectedly, we find sterol accumulating at the TGN rather than lysosomes in GARP-deficient neurons. Altering the transport or availability of sterol and related lipids at the TGN modulates GARP null phenotypes. In particular, overexpressing *oxysterol binding protein (Osbp)* or knocking down the PI4P kinase, *four wheel drive (fwd)*, exacerbates the dendrite regrowth phenotype in $Vps54^{KO/KO}$ neurons, while haploinsufficiency of *Osbp* rescues it.

## Results

### Reduced lifespan of GARP knockout flies

In mice, homozygous null mutants of *Vps52* and *Vps54* are lethal at early embryonic stages (Sugimoto et al., 2012; Schmitt-John

et al., 2005), limiting the study of the GARP and EARP complexes in these models. To overcome these challenges, we made use of the genetic toolbox available in *Drosophila*. To study the role of the GARP and EARP complexes, we generated knockouts (KO) of the EARP-specific *Vps50*, the shared core component *Vps53*, and the GARP-specific *Vps54* (also known as *scattered* in flies), using CRISPR/Cas9 gene editing to replace the entire coding sequence of each gene (see Fig. S1 for schematic and genotyping). We confirmed by RT-PCR that expression of each gene targeted for KO was eliminated (Fig. 1 B). While we could not find antibodies to determine Vps50 or Vps53 protein levels in *Drosophila*, we were able to confirm that Vps54 protein was absent from $Vps54^{KO/KO}$ larvae (Fig. S1 H).

In flies, homozygous knockouts of *Vps53* ($Vps53^{KO/KO}$) survived the larval stages but died during pupation. Ubiquitous expression of UAS-Vps53 using either *tubulin*-Gal4 or *daughterless*-Gal4 in $Vps53^{KO/KO}$ mutants allowed for survival to adulthood. In contrast to mice, homozygous knockout flies of the complex-specific components, $Vps50^{KO/KO}$ or $Vps54^{KO/KO}$, were viable to adulthood. Loss of the GARP-specific Vps54 ($Vps54^{KO/KO}$) reduced lifespan in both males and females (Fig. 1, C–F). Control males lived an average of 46.7 ± 0.9 d and a maximum of 82 d, whereas $Vps54^{KO/KO}$ male flies lived on average only 24.6 ± 0.6 d and a maximum of 46 d. Control female flies lived an average of 47.6 ± 1.2 d and a maximum age of 82 d, whereas $Vps54^{KO/KO}$ females lived an average of only 23.7 ± 0.6 d and a maximum age of 49 d. We generated hemizygotes by crossing $Vps54^{KO}$ flies to a chromosomal deficiency (*Df*[2L] *Exel8022*) and confirmed that they also had shortened lifespan. Loss of the EARP-specific component *Vps50* did not consistently reduce lifespan across genotypes.

The GARP complex has also been implicated in spermiogenesis in both mice and flies (Schmitt-John et al., 2005; Castrillon et al., 1993). In fact, the name for the *Drosophila* homolog of Vps54, *scattered*, refers to the disorganized, scattered organization of nuclei in developing spermatids. To further characterize these knockouts, we therefore also tested fertility of male files. $Vps54^{KO/KO}$ and $Vps54^{KO}/Df$ males, like the $scat^{1/1}$ null males, were sterile. In contrast, $Vps50^{KO/KO}$ and $Vps50^{KO}/Df$ males were fertile.

**Loss of either the EARP or GARP complex impairs arborization of adult neurons**

To determine how knockout of the EARP and GARP complexes may affect neuron development, we first examined the overall morphology of c4da neurons in adult pharate flies. To circumvent difficulties owing to the adult lethality of $Vps53^{KO/KO}$, we used MARCM (mosaic analysis with a repressible cell marker; Lee and Luo, 1999) to generate homozygous knockout clonal neurons in the viable heterozygous flies, to evaluate the role of *Vps53* in neuronal morphology. Dendrite arbors in $Vps53^{KO/KO}$ clones were only about a third of the total length of controls (Fig. 2, A and B), revealing a cell autonomous requirement of *Vps53* for dendritic morphology. The arbors of $Vps53^{KO/KO}$ clones were also less complex and contained fewer total branches (Fig. 2, C and D). The cell-autonomous involvement of *Vps53* was validated by showing that the dendrite branch length and

number in the $Vps53^{KO/KO}$ clones can be rescued using the class IV specific *ppk*-Gal4 to drive expression of wild-type Vps53 protein.

Given that loss of Vps53 disrupts both the EARP and GARP complexes, we next analyzed neuronal morphology in knockouts targeting the complex-specific components, *Vps50* and *Vps54*, respectively (Fig. 3). Whole-body knockout of either *Vps50* or *Vps54* reduced the total dendritic length and branch number in adult neurons as compared to controls (Fig 3, A–D). For both parameters, the GARP-specific $Vps54^{KO/KO}$ had a stronger effect than $Vps50^{KO}$. These dendrite morphology defects can be rescued by expression of the respective wild-type protein in neurons through the *ppk*-Gal4, suggesting that the GARP and EARP complexes function cell autonomously to regulate dendrite morphogenesis. RNAi knockdown of EARP and GARP complex components in c4da neurons further confirmed the cell-autonomous function of these complexes on adult dendrite arborization (Fig. S2 A).

While c4da neurons establish dendritic arbors initially in the larval stage, the dendrites are extensively pruned in early pupation and these neurons subsequently regrow a remodeled adult arbor (Kuo et al., 2005; Williams and Truman, 2005; Shimono et al., 2009). To gain a better understanding of when the EARP and GARP complexes are required during this dynamic period of development, we analyzed dendrite morphology in both larvae and pupae. The c4da neurons in $Vps50^{KO/KO}$, $Vps53^{KO/KO}$, or $Vps54^{KO/KO}$ larvae grew dendritic arbors comparable in length to controls (Fig. S2 B). Given that the c4da neurons must scale in size to keep pace with larval growth and maintain coverage of their receptive fields (Parrish et al., 2009), we also measured the coverage index (neuron area/receptive field area) and again observed no difference between control and knockout neurons (Fig. S2 C). There also does not appear to be any significant effect of maternally contributed Vsp50 or Vps54 to dendrite growth, as neurons in larvae from homozygous knockout mothers grew arbors comparable in size to controls. We therefore concluded that the GARP and EARP complexes are dispensable for larval neuron growth.

Differences between control and knockout arbors first appeared as the dendrites regrew their adult arbors (Fig. 3 E). The phenotype in $Vps54^{KO/KO}$ neurons emerged just at the end of pupation by 96 h after puparium formation (APF), while the $Vps50^{KO/KO}$ phenotype emerged slightly later in 1-d-old adults. To better characterize the dynamics of dendrite regrowth during this period, we analyzed dendrite length across developmental timepoints within each genotype. Wild-type neurons grew between 72 and 96 h APF, paused during eclosion, and grew again between 1 and 7 d in the pharate adults (Fig. S2 D). We did not observe any additional dendrite growth beyond 7 d. $Vps50^{KO/KO}$ neurons also grew during those same periods (Fig. S2 E), however to a lesser extent. $Vps54^{KO/KO}$ neurons grew slightly between 72 and 96 h APF but did not continue to grow after eclosion (Fig. S2 F). We conclude that loss of either the EARP or GARP complexes does not result in a delay of dendrite regrowth, but rather decreases the magnitude of regrowth.

C4da neurons project to the abdominal neuromere of the ventral nerve cord (VNC), where they synapse with second-

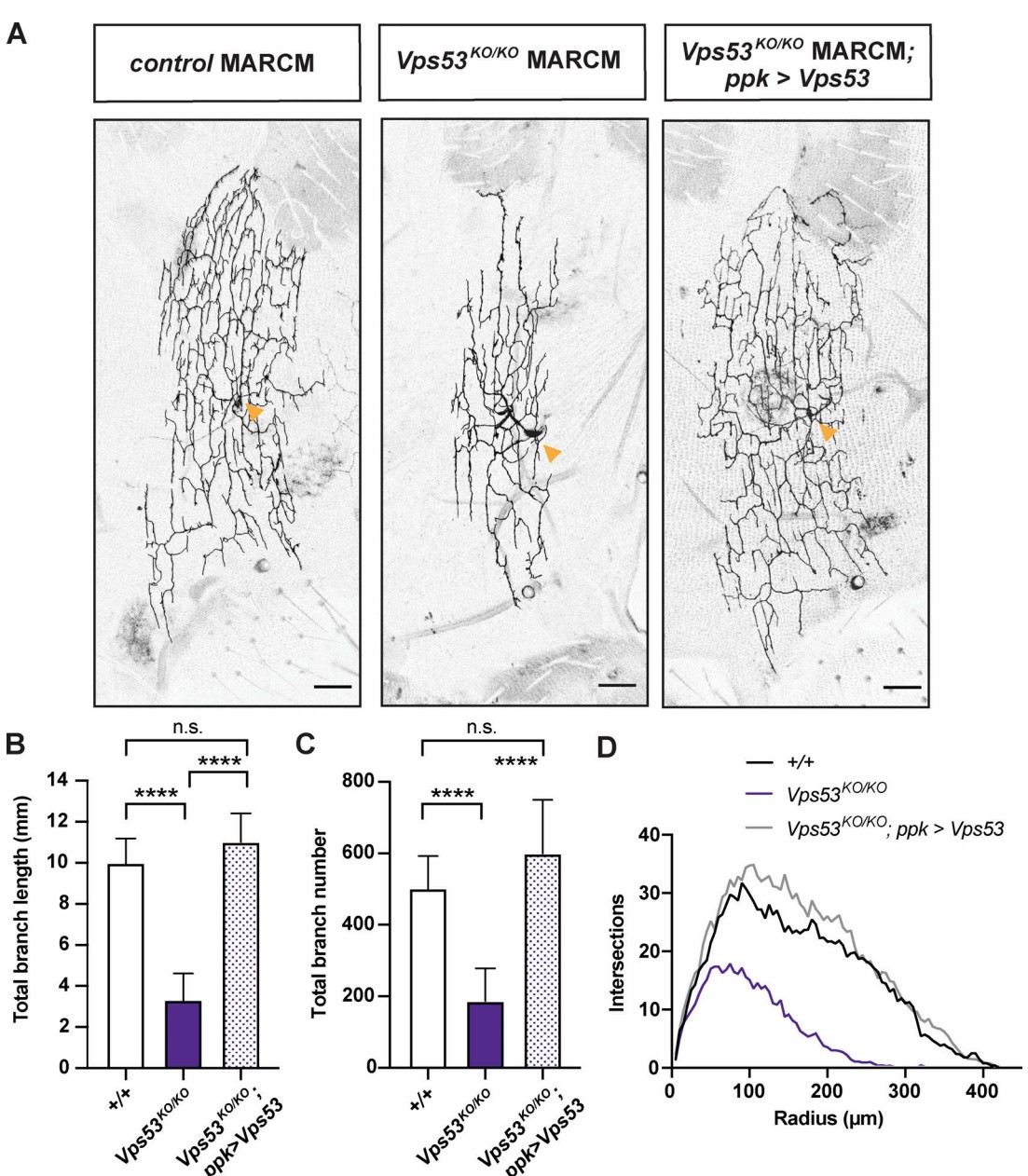

**Figure 2. Vps53$^{KO/KO}$ MARCM clonal neurons have smaller dendritic arbors. (A)** Representative maximum z-projections of MARCM control FRT$^{40A}$, Vps53$^{KO/KO}$, and Vps53$^{KO/KO}$; ppk > Vps53 c4da neuron clones labeled with ppk-Gal4, UAS-CD4-tdGFP. Images were collected from 7-d-old male pharate adults. Yellow arrows point to the soma. Scale bar = 50 μm. **(B)** Quantification of total dendrite branch length, ****P < 0.0001, n.s. P = 0.1797. **(C)** Quantification of total branch number, ****P < 0.0001, n.s. P = 0.2579. Both total dendrite branch length and number were analyzed by one-way ANOVA with Tukey's post-test. Data presented as mean ± SD. **(D)** Sholl analysis. Curves are the average number of intersections/radius. For B–D, MARCM control FRT$^{40A}$ n = 8 independent neurons; Vps53$^{KO/KO}$ n = 13 independent neurons; and Vps53$^{KO/KO}$; ppk > Vps53 n = 11 independent neurons. Flies were collected from at least three independent experiments.

order sensory neurons (Tsubouchi et al., 2017; Court et al., 2020). To determine whether loss of the GARP and EARP complexes specifically affects dendrites or if it may impact overall neuronal morphology, we also examined axon terminal morphology in c4da neuron MARCM clones. We did not observe any differences in c4da axon terminal branch length in Vps53$^{KO/KO}$ or Vps54$^{KO/KO}$ clones (Fig. S2, G and H), suggesting that the dendritic phenotype is specifically due to a defect in dendrite regrowth and not a secondary effect of axon retraction.

Class I da neurons (c1da) also remodel their dendrites from a relatively simple larval arbor to a more complex arbor during metamorphosis (Shimono et al., 2009). Similar to our observations of c4da neurons, c1da neurons in Vps50$^{KO/KO}$ or Vps54$^{KO/KO}$ larvae were comparable to wild-type controls (Fig. 4 C). After pruning and regrowth, Vps54$^{KO/KO}$ but not Vps50$^{KO/KO}$ c1da neurons in adults had reduced dendrite branch length (Fig. 4, A and B), indicating that the GARP complex is required in multiple neuron types during developmental remodeling.

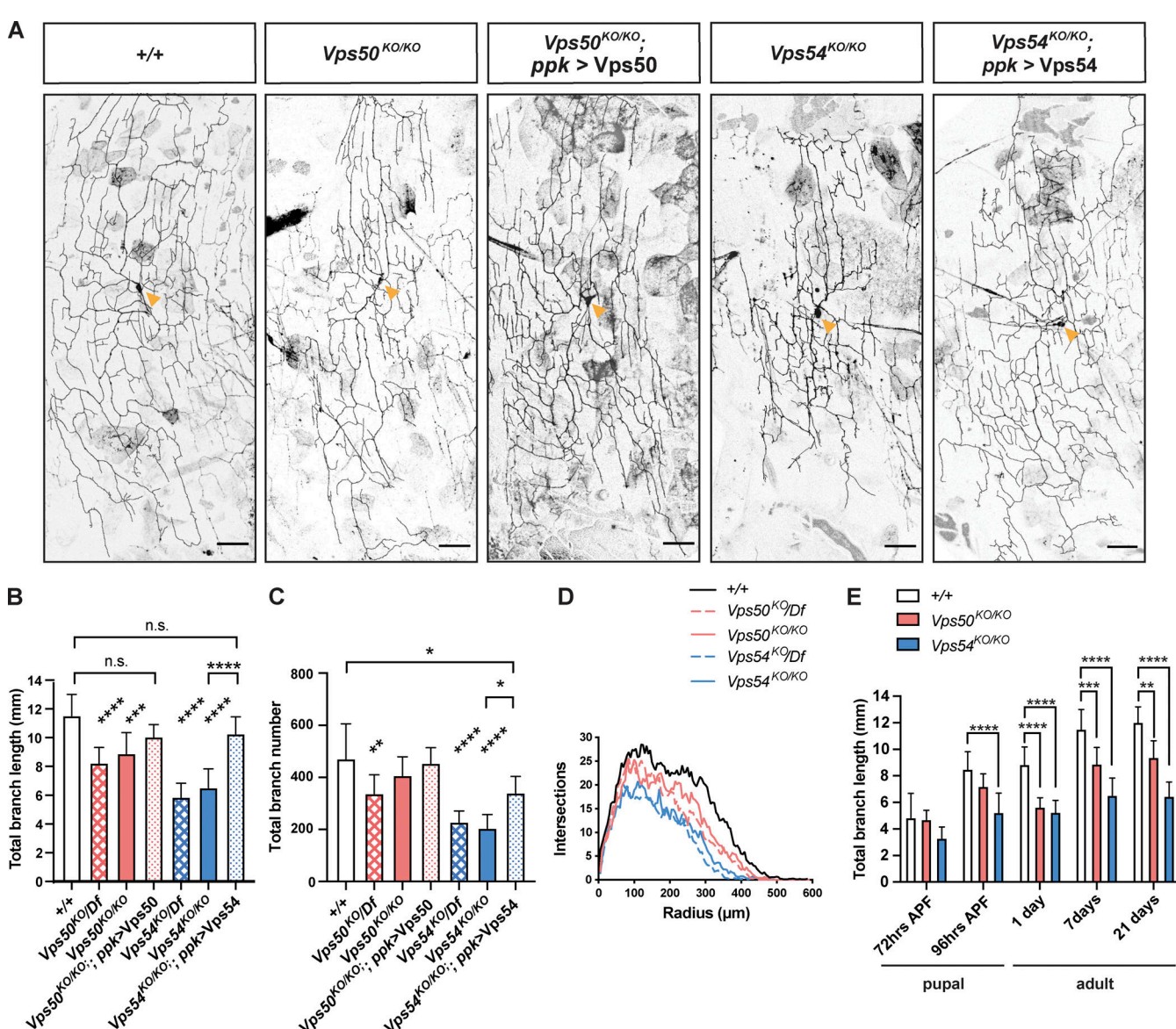

Figure 3. **Both the GARP and EARP complexes are necessary for dendrite arborization in class 4 da neurons. (A)** Representative maximum intensity z-projections of class IV da neurons labeled with ppk-Gal4, UAS-CD4-tdTomato from 7-d-old male pharate adults. Yellow arrows point to the soma. Scale bar = 50 µm. Both total dendrite branch length and number were analyzed by one-way ANOVA with Tukey's post-test. Data presented as mean ± SD. **(B)** Quantification of total dendrite length ***P = 0.0001, ****P < 0.0001, +/+ vs. $Vps50^{KO/KO}$; $ppk > Vps50$, P = 0.278, +/+ vs. $Vps54^{KO/KO}$; $ppk > Vps54$, P = 0.3165m, $Vps50^{KO/KO}$ vs. $Vps54^{KO/KO}$ ***P = 0.0003. **(C)** Quantification of total dendrite branch number. +/+ vs. $Vps54^{KO/KO}$; $ppk > Vps54$, *P = 0.0107, $Vps54^{KO/KO}$ vs. $Vps54^{KO/KO}$; $ppk > Vps54$, *P = 0.0178), +/+ vs. $Vps50^{KO/KO}$ **P = 0.009, ****P < 0.0001, $Vps50^{KO/KO}$ vs. $Vps54^{KO/KO}$ ****P < 0.0001, remaining n.s. comparisons P > 0.27. **(D)** Sholl analysis. Curves are the average number of intersections/radius. For B–D, n = 7–12 independent neurons/genotype. **(E)** Quantification of total dendrite branch length over development from 72 h APF to 21 d after eclosion. Analyzed by two-way ANOVA with Tukey's post-test. Data presented as mean ± SD.**P < 0.01, ***P < 0.001, ****P < 0.0001. Please see Fig. S2, D–F for additional statistics and Table S2 for the full list of P values for the comparisons in E. +/+ n = 10 independent neurons/timepoint; $Vps50^{KO/KO}$ n = 9–11 independent neurons/timepoint; $Vps54^{KO/KO}$ n = 10–12 independent neurons/timepoint. For B–E, flies were collected from at least three independent experiments.

## Complex-specific defects in secretory and endosomal organelles

Given the role of the EARP and GARP complexes in regulating specific steps in membrane trafficking, we examined various markers of the endolysosomal and secretory pathways in 1-d-old adults, when knockouts of both complexes exhibited dendrite phenotypes. The number of Rab5+ early endosomes was increased in the soma of $Vps50^{KO/KO}$ but not $Vps54^{KO/KO}$ neurons (Fig. 5, A and B), consistent with the fact that the EARP complex

facilitates cargo sorting from early endosomes to Rab4+ recycling endosomes (Schindler et al., 2015). We did not observe changes in early endosome area or mean fluorescence intensity of the Rab5 staining (Fig. S3, A and B), nor did we observe changes in Rab5 staining in proximal dendrites (Fig. S3, C–E). The number of Rab7+ late endosomes was increased in the soma of $Vps54^{KO/KO}$ but not $Vps50^{KO/KO}$ neurons (Fig. 5, C and D; and Fig. S3, F and G), indicative of complex-specific defects in endosome populations. Dendrites are devoid of degradative lysosomes, and therefore

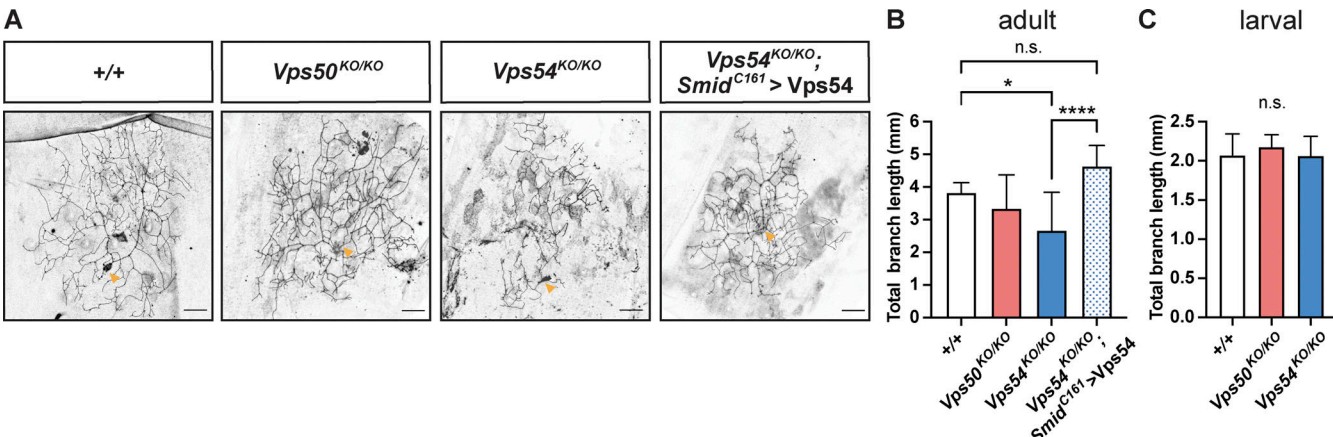

Figure 4. **Loss of the GARP complex also impairs class I da neuron dendrite remodeling. (A)** Representative maximum intensity z-projections of c1da neurons from +/+ (UAS-CD4-tdTomato/+; $Smid^{C161}$ Gal4/+), $Vps50^{KO/KO}$ ($Vps50^{KO}$, UAS-CD4-tdTomato/$Vps50^{KO}$; $Smid^{C161}$ Gal4/+), $Vps54^{KO/KO}$ ($Vps54^{KO}$, UAS-CD4-tdTomato/$Vps54^{KO}$; $Smid^{C161}$ Gal4/+), and $Vps54^{KO/KO}$, Smid161>Vps54 ($Vps54^{KO}$, UAS-CD4-tdTomato/$Vps54^{KO}$; $Smid^{C161}$ Gal4/UAS-Vps54) pharate adults (0–8 h after eclosion). Yellow arrows point to the soma. Scale bar = 25 µm. **(B)** Quantification of total dendrite length of adult c1da neurons. *P = 0.0222, ****P < 0.0001, n.s. P > 0.16. **(C)** Quantification of larval c1da neurons (96 h AEL) labeled with Gal4$^{221}$>UAS-CD4-tdGFP, all comparisons n.s. P > 0.78. Both B and C were analyzed by one-way ANOVA with Tukey's post-test. Data presented as mean ± SD. N = 10 independent neurons/genotype. Samples were collected from at least three independent experiments.

endosomal cargo destined for degradation must be trafficked to the soma (Yap et al., 2018). We did not observe any significant change in Rab7+ positive endosomes in the dendrites of $Vps54^{KO/KO}$ neurons (Fig. S3, H–J), suggesting their trafficking was not affected. These changes in endosomal populations did not occur in larval neurons (Fig. S3, P and Q), supporting the notion that these complexes are dispensable for larval neurodevelopment.

We also examined the lysosomal markers GFP-Lamp (Fig. 5, E and F; and Fig. S3, K and L) and Spinster-RFP (Spin-RFP; Dermaut et al., 2005; Rong et al., 2011; Fig. S3, M–O) and observed an increase in the number and size of organelles positive for these markers, respectively, in the soma of $Vps54^{KO/KO}$, but not $Vps50^{KO/KO}$ neurons. Lysosomal expansion can be a result of impaired cargo degradation by the resident acid hydrolases such as cathepsins. Immature forms of acid hydrolases are trafficked from the secretory pathway to lysosomes in a GARP-dependent manner (Pérez-Victoria et al., 2008). Upon reaching the acidic environment of the lysosome, hydrolases are processed into their mature, active forms. Therefore, we also examined the processing of cathepsin L (catL) by Western blot in head lysates. We did not observe any difference in catL processing in young adult flies in $Vps54^{KO/KO}$ or $Vps50^{KO/KO}$ neurons (Fig. S3, R–T), suggesting that catL is successfully trafficked to acidic lysosomes in both knockout lines. Further, these results suggest that the inability of neurons to regrow their adult arbors during pupation may be independent of their lysosomal degradative capacity.

Because the GARP complex regulates retrograde traffic to the TGN, we also examined this compartment by staining for Golgin245. The number of Golgin245 puncta was increased in the soma of $Vps54^{KO/KO}$ but not $Vps50^{KO/KO}$ neurons (Fig. 5, G and H; and Fig S3, U–Y). If the increase in puncta number were due to fragmentation of the Golgi, we would expect the puncta to be smaller in size. However, we did not observe a significant difference in the size of Golgin245 puncta (Fig. S3 U), suggesting the increase in puncta number is not a result of Golgi fragmentation.

### Sterol accumulates at the TGN rather than endolysosomes in GARP KO neurons

Previous studies have reported accumulation of sterols in cells lacking the GARP complex (Wei et al., 2017; Fröhlich et al., 2015). We therefore sought to examine sterol levels and localization in knockout neurons. Filipin is a widely used fluorescent stain that binds to free sterols. $Vps54^{KO/KO}$ neurons exhibited strong filipin staining in the soma compared to controls, which was rescued by expression of wild-type Vps54 (Fig. 6, A and B). Sterol accumulation in GARP deficient neurons appeared to be transient and to correlate with the emergence of the dendrite morphology defect in $Vps54^{KO/KO}$ neurons at 96 h APF (Fig. 6 C). $Vps50^{KO/KO}$ neurons, however, showed filipin staining comparable to controls throughout development (Fig. 6 C), indicating that the GARP, but not EARP complex, plays a role in sterol processing.

A previous study in mammalian cells determined that sterols accumulate in the late endosomal/lysosomal compartment in Vps52-deficient cells due to missorting of NPC2 (Wei et al., 2017). Surprisingly, we did not observe any significant accumulation of sterols in endolysosomes labeled with either Rab7 (Fig. 6, D and E) or Spin-RFP (Fig. S4 A). While the strongest internal filipin signal appeared to occur in the ER, as indicated by the marker Sec61β, we did not find any differences in filipin intensity between genotypes in this organelle (Fig. S4, C and D). Of the organelle markers we examined, we only found a significant increase in filipin staining in the Golgin245-positive compartment corresponding to the TGN (Fig. 6, F and G). It thus appears that in GARP-deficient neurons, sterols can exit the endolysosomal pathway but aberrantly accumulate in the secretory pathway instead.

### Targeting-specific lipid regulators at the TGN modulates GARP KO phenotypes

To gain a better understanding of how sterols may accumulate at the TGN in the $Vps54^{KO/KO}$, we examined genetic interactions

JCB

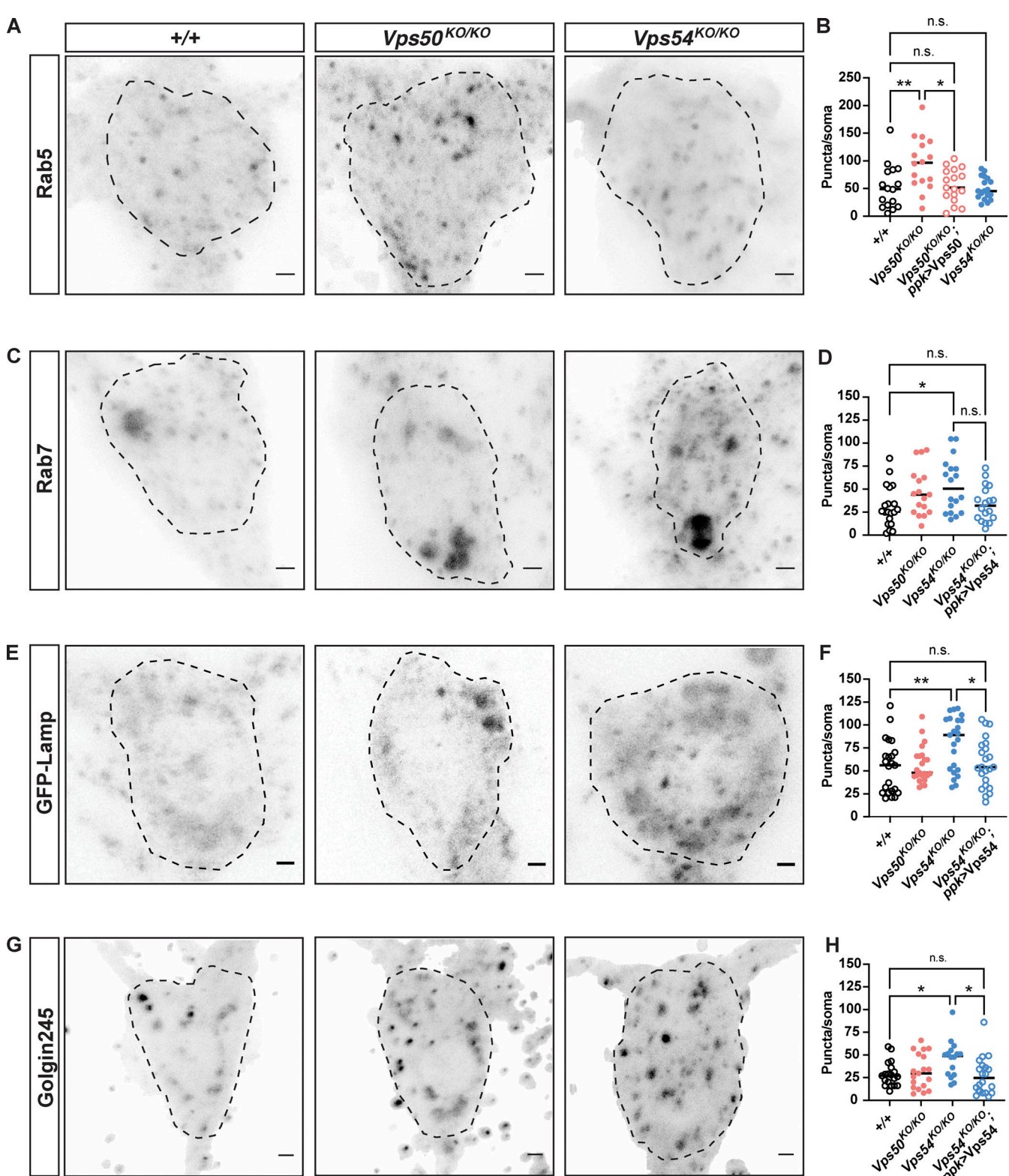

Figure 5. **Complex-specific impairments in organelle populations. (A)** Maximum intensity z-projections of endogenous Rab5 staining in neurons from 1-d-old flies. Dashed lines indicate soma area. Scale bars = 2.5 µm. **(B)** Quantification of the number of Rab5 puncta/soma, n = 16–18 independent samples/genotype. For all graphs in this figure, each data point represents the average of 1–3 cells/sample. Data for each organelle marker was obtained from at least three independent experiments. All organelle data were analyzed by one-way ANOVA with Tukey's post-test. +/+ vs. *Vps50*^KO/KO **P = 0.0061, *Vps50*^KO/KO vs. *Vps50*^KO/KO; *ppk* > Vps50 *P = 0.0106. All other comparisons, n.s. P > 0.96. **(C)** Maximum intensity z-projections of endogenous Rab7 staining. **(D)** Quantification of the number of Rab7 puncta/soma, n = 17–20 independent samples/genotype. +/+ vs. *Vps50*^KO/KO *P = 0.0251, *Vps54*^KO/KO vs. *Vps54*^KO/KO; *ppk* > Vps54 n.s. P = 0.0746. All other comparisons n.s. P > 0.37. **(E)** Maximum intensity z-projections of the transgene UAS-GFP-Lamp expressed in c4da neurons from +/+ (*ppk* > GFP-Lamp), *Vps50*^KO/KO (*Vps50*^KO, UAS-GFP-Lamp/*Vps50*^KO; *ppk*-Gal4, UAS-CD4-tdTomato/+), *Vps54*^KO/KO (*Vps54*^KO, UAS-GFP-Lamp/*Vps54*^KO; *ppk*-Gal4, UAS-

CD4-tdTomato/+), and *Vps54^{KO/KO}* (*Vps54^{KO}*, UAS-GFP-Lamp/*Vps54^{KO}*; *ppk*-Gal4, UAS-CD4-tdTomato/UAS-Vps54). **(F)** Quantification of the number of GFP-Lamp puncta/soma, *n* = 23 independent samples/genotype. +/+ vs. *Vps54^{KO/KO}* **P = 0.006, *Vps54^{KO/KO}* vs. *Vps54^{KO/KO}*; *ppk* > Vps54 *P = 0.0268. All other comparisons n.s. P > 0.99. **(G)** Maximum intensity z-projection of endogenous Golgin245 staining. **(H)** Quantification of the number Golgi245 puncta/soma, *n* = 18–21 independent samples/genotype. +/+ vs. *Vps54^{KO/KO}* *P = 0.0425. *Vps54^{KO/KO}* vs. *Vps54^{KO/KO}*; *ppk* > Vps54 *P = 0.0173. All other comparisons n.s. P > 0.12. For additional organelle parameters, see Fig. S3.

between *Vps54* and various sterol and lipid regulatory proteins. Oxysterol binding protein (Osbp) regulates transport of sterols across several interorganelle contact sites. At contacts between the ER and TGN, Osbp interacts with the ER-localized protein VAP-A to facilitate the transfer of sterol from the ER in exchange for PI4P (Mesmin et al., 2017). We therefore made crosses to bring either a null Osbp allele (*Osbp^1*; Ma et al., 2010) or UAS-Osbp into the *Vps54^{KO/KO}* background. Removal of one functional *Osbp* allele (*Vps54^{KO/KO}*; *Osbp^{1/+}*) rescued the dendrite morphology defect in *Vps54^{KO/KO}* neurons, while Osbp overexpression (*Vps54^{KO/KO}*; *ppk* > Osbp) dramatically exacerbated it (Fig. 7, A and B). To evaluate the contribution of Osbp acting at TGN/ER contact sites, we next targeted its binding partner, the single *Drosophila* homolog of VAP-A/B, *Vap33* (Pennetta et al., 2002). Neither knockdown nor overexpression of Vap33 had any effect on the *Vps54^{KO/KO}* phenotype (Fig. S5). We next targeted the PI4-kinase that phosphorylates phosphatidylinositol to generate PI4P, known as *fwd* (Polevoy et al., 2009) in *Drosophila*. RNAi knockdown of *fwd* itself decreased the total dendrite branch length in c4da neurons (Fig. 7, C and D). We reasoned that if Osbp-mediated exchange of sterol for PI4P between the ER and TGN was responsible for the accumulation of sterol at the TGN in *Vps54^{KO/KO}* neurons, then *fwd* knockdown should rescue the *Vps54^{KO/KO}* dendrite morphology defect. However, expressing *fwd* RNAi in *Vps54^{KO/KO}* neurons (*Vps54^{KO/KO}*; *fwd* RNAi) exacerbated the *Vps54^{KO/KO}* phenotype (Fig. 7, C and D). Taken together, these results suggest that TGN/ER contacts are unlikely the source for sterol accumulation at the TGN in *Vps54^{KO/KO}* neurons.

To gain additional insight into the contribution of Osbp and fwd to these phenotypes, we also examined relevant lipids at the TGN. While filipin staining in neurons from the *Osbp^{1/+}* mutant alone was comparable to controls, this null allele rescued both the TGN-associated and total filipin staining in *Vps54^{KO/KO}* neurons (*Vps54^{KO/KO}* vs. *Vps54^{KO/KO}*; *Osbp^{1/+}*; Fig. 7, E and F). Unexpectedly, overexpression of Osbp in the *Vps54^{KO/KO}* background also decreased both total and TGN-associated filipin levels, despite exacerbating the dendritic phenotype (Fig. 7, E and F). When *fwd* was knocked down (*Vps54^{KO/KO}*; *fwd* RNAi), total and TGN-associated filipin levels remained elevated and were not significantly different from those in *Vps54^{KO/KO}* neurons (Fig. 7, G and H). Because Osbp exchanges sterol for PI4P at TGN/ER contacts, we also examined PI4P levels using the reporter P4M-GFP, which binds specifically to PI4P in cell membranes (Balakrishnan et al., 2018). In control cells, we observe a strong P4M-GFP signal at the TGN (Fig. 7 I). As expected, expression of *fwd* RNAi decreased TGN levels of P4M-GFP (Fig. 7, I and J), validating the effectiveness of this RNAi. However, we did not see any effect of either *Vps54^{KO/KO}* or *Osbp^{1/+}* on P4M-GFP at the TGN (Fig. 7 K). These data together suggest that Osbp-dependent

sterol transfer site(s) other than the sterol/PI4P exchange cycle between the TGN and ER must contribute to the elevated sterol levels at the TGN in *Vps54^{KO/KO}* neurons.

To further investigate the ability of a single *Osbp^1* allele to rescue the *Vps54^{KO/KO}* phenotypes, we also examined organelle morphology. *Osbp^1* heterozygosity rescued the number of Golgin245 puncta (Fig. 8, A–D) but not the number of Rab7+ late endosomes in *Vps54^{KO/KO}* neurons (*Vps54^{KO/KO}*; *Osbp^{1/+}*; Fig. 8, E–H). These results further support the notion that the inability of dendrites to regrow in *Vps54^{KO/KO}* neurons is due to perturbations at the TGN, and not to impaired endolysosomal trafficking.

## Discussion

### GARP and EARP in neurodevelopment

The links of the GARP and EARP complexes to neurodevelopmental disease notwithstanding, our understanding of how these complexes function in neurons remains limited. Studies of these complexes have been hampered by the early embryonic lethality of mice lacking components of these complexes (Sugimoto et al., 2012; Schmitt-John et al., 2005). In our *Drosophila* mutant studies, we show that *Vps50*, *Vps53*, or *Vps54* is dispensable for larval development. Loss of *Vps53* resulted in pupal lethality, while *Vps54* knockouts had a reduced lifespan as adults. The GARP complex is required for dendrite regrowth in both c4da and c1da neurons after developmental pruning, while the EARP complex is only required in c4da neurons. The emergence of neuronal phenotypes only at later developmental stages is reminiscent of the secondary microcephaly that emerges postnatally in patients with GARP/EARP complex mutations (Gershlick et al., 2018; Feinstein et al., 2014; Hady-Cohen et al., 2018). There are several explanations for why an EARP-dependent phenotype was not observed in c1da neurons. It is possible that Vps50 is not highly expressed in c1da neurons. Because c1da neurons undergo developmentally programmed cell death within the first few days of adulthood (Shimono et al., 2009), we examined their morphology within a few hours of eclosion to avoid any loss of neurons. It is possible that a phenotype in the *Vps50^{KO/KO}* flies might emerge at a slightly later timepoint prior to c1da apoptosis.

We did not detect morphological changes in axon terminals of *Vps53^{KO/KO}* or *Vps54^{KO/KO}* c4da neurons in 7-d-old adults, but we cannot rule out later degeneration that may occur in these knockouts. Because Vps53 is found in both the GARP and EARP complexes, we conclude that neither complex is required for development of c4da axon terminals. However, due to reagent limitations, we cannot conclude whether *Vps50*, independently of the EARP complex, is required for axon morphology. Given that we observed a reduced lifespan in the *Vps54^{KO/KO}* flies, it will

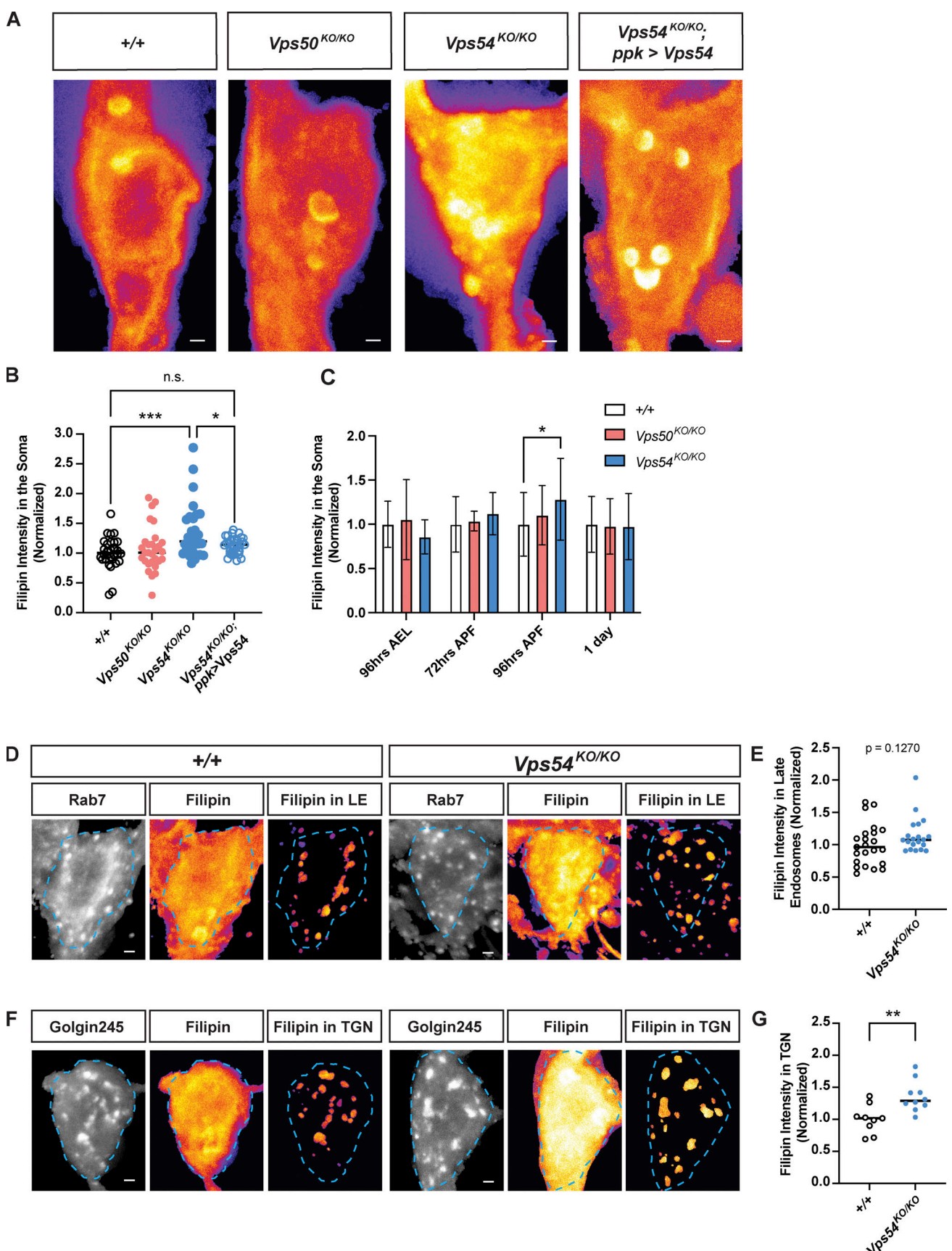

Figure 6. **Accumulation of free sterol at the TGN during dendrite regrowth in GARP deficient neurons. (A)** Maximum intensity z-projections showing total filipin staining in the soma of *+/+*, *Vps50^{KO/KO}*, *Vps54^{KO/KO}*, and *Vps54^{KO/KO}; ppk > Vps54* neurons at 96 h APF. Scale bar = 1 μm. Images are pseudocolored

with the fire LUT in which cooler colors indicate lower and hotter colors indicate higher fluorescence intensity values. Bright spots within the soma appear to overlap with the ER marker Sec61β (see Fig. S4, C and D). **(B)** Quantification of filipin fluorescence intensity at 96 h APF, $n$ = 30–33 independent samples/genotype. Data were normalized to average control value for each experiment to account for inter-experimental differences in filipin intensity. Analyzed by one-way ANOVA with Tukey's post-test. +/+ vs. $Vps54^{KO/KO}$ ***P = 0.0004, $Vps54^{KO/KO}$ vs. $Vps54^{KO/KO}$; $ppk$ > Vps54 *P = 0.0463. Other comparisons n.s. P > 0.45. **(C)** Quantification of filipin fluorescence intensity in +/+, $Vps50^{KO/KO}$, $Vps54^{KO/KO}$ neurons across development. $N \geq 17$ independent samples/genotype at each timepoint. Analyzed by two-way ANOVA with Šidák's multiple comparison's correction, *P = 0.0312. **(D–G)** Sterol levels in organelles. **(D)** For each genotype, left and middle panels show maximum intensity projections of Rab7 and filipin staining. To obtain the filipin signal from late endosomes, a mask was generated using the Rab7 z-stack and applied to the filipin z-stack. Right panel shows maximum intensity projection of the extracted filipin signal in Rab7+-late endosomes. Dashed line shows soma area. **(E)** Quantification of late endosome-associated filipin staining. Analyzed by two-sided Mann-Whitney U test, n.s. P = 0.1229. $N$ = 20–23 independent samples/genotype. **(F)** As in D, except with Golgin245 to obtain filipin signal in the TGN. **(G)** Quantification of TGN-associated filipin signal. Analyzed by unpaired two-sided $t$ test, **P = 0.0038. $N$ = 9–10 independent samples/genotype. Samples were collected from at least three independent experiments.

be of interest to examine in future studies whether age-dependent changes in neuronal morphology or function may occur in addition to the developmental phenotypes characterized in this study. Studies in the wobbler mouse, bearing a spontaneous point mutation in $Vps54$, reveal the degeneration of multiple brain regions and motor neurons in adult mice (Schmitt-John et al., 2005; Schmitt-John, 2015). The motor neuron phenotype in mice is distinct from that observed in $Drosophila$ mutants harboring the $Vps54$ null allele, $scat^1$, which exhibit overgrowth of the larval neuromuscular junction (Patel et al., 2020). Taken together with our study, these findings suggest that loss of the GARP/EARP complexes differentially affects distinct neuronal populations across lifespan.

## Appearance of endolysosomal phenotypes only in later developmental stages

At the subcellular level, loss of either the GARP or EARP complexes results in distinct effects on the endolysosomal system. Knockout of $Vps50$ specifically affects early endosomes, while knockout of $Vps54$ specifically affects late endosomes and lysosomes. Previous studies have shown that the GARP complex is essential for the proper sorting of lysosomal hydrolases. Despite an enlargement of the lysosomal population, we detected no changes in expression or maturation of the hydrolase cathepsin L in head lysates from 1-d-old knockout flies. This is consistent with a report on the retromer complex, which functions upstream of the GARP complex in hydrolase sorting. In that study (Ye et al., 2020), changes in cathepsin L processing are observed in 30 d old, but not 1 d old, $Vps29$ mutant flies. The authors of that study suggest that there must be compensatory mechanisms that facilitate proper lysosomal hydrolase sorting during earlier stages of development. Our results showing that cathepsin L processing is intact in young adult flies, as well as the dispensable role of the GARP/EARP complexes for overall larval development, further support this notion.

## GARP in sterol transport in neurons

In $Saccharomyces cerevisiae$, $Vps53\Delta$ or $Vps54\Delta$ cells accumulate sterol intracellularly (Fröhlich et al., 2015). As yeast lack the EARP complex, this was assumed to be a function of the GARP complex. In mammalian cells, knockdown of the shared component Vps52 results in mis-sorting of NPC2 (Wei et al., 2017), leading to sterol accumulation in endolysosomes. This study, however, did not target the complex-specific components of the

GARP and EARP complexes. We find that sterol accumulates in neurons of $Vps54^{KO/KO}$ but not $Vps50^{KO/KO}$ $Drosophila$, suggesting that the EARP complex may not be involved sterol transport. To our surprise, we observed accumulation of sterol in $Vps54^{KO/KO}$ neurons at the TGN, not in late endosomes or lysosomes. Osbp likely facilitates sterol transport to the TGN in $Vps54^{KO/KO}$ neurons, as the $Osbp^{1/+}$ null ($Vps54^{KO/KO}$; $Osbp^{1/+}$) rescued sterol levels, TGN morphology, and the dendritic phenotype of GARP-deficient neurons. Strikingly, $Osbp^{1/+}$ heterozygosity did not rescue the observed defects in endolysosomal morphology, indicating that these changes are still permissive to dendrite regrowth. Taken together with the data showing no impairment of cathepsin L maturation in $Vps54^{KO/KO}$ lysates, these results suggest that perturbed dynamics at the TGN, but not in endolysosomes, contributes in part to impaired dendrite regrowth.

Our results showing that overexpression of Osbp in $Vps54^{KO/KO}$ neurons also decreased filipin levels at the TGN while exacerbating the dendritic phenotype suggests that this manipulation may disturb the balance of sterol transport at several interorganelle contact sites. For example, it is possible that the decrease in TGN-associated filipin staining upon Osbp overexpression may be an indirect effect of increased transport of sterol out of the secretory pathway through ER-endolysosome contacts (Dong et al., 2016). Additionally, Osbp functions that are independent of sterol transport may influence dendrite morphology. This is supported by our data showing that the overexpression of Osbp in the wild-type background decreased total dendrite length without affecting sterol levels. In this context, Osbp overexpression may alter signaling pathways that act in parallel to regulate dendrite morphology. For example, Osbp creates a scaffold for protein phosphatases, including protein phosphatase 2a (PP2A; Wang, 2005), which is essential for dendrite pruning and cytoskeletal dynamics in c4da neurons (Rui et al., 2020; Wolterhoff et al., 2020).

Because Osbp regulates sterol transport through multiple interorganelle contact sites, further study is required to identify the precise sites involved in the transport of sterol to the TGN in GARP-deficient neurons. Our genetic interaction studies with $Vap33$ indicate that interorganelle contact sites other than the ER-TGN contact sites mediated by Osbp may be responsible for the accumulation of sterol at the TGN. One site of interest is the TGN-Rab11+ recycling endosome contact site. At these sites, Osbp binds the Rab11 interacting protein RELCH (Sobajima et al., 2018). This study demonstrated that knockdown of Osbp, Rab11, or RELCH decreased sterol

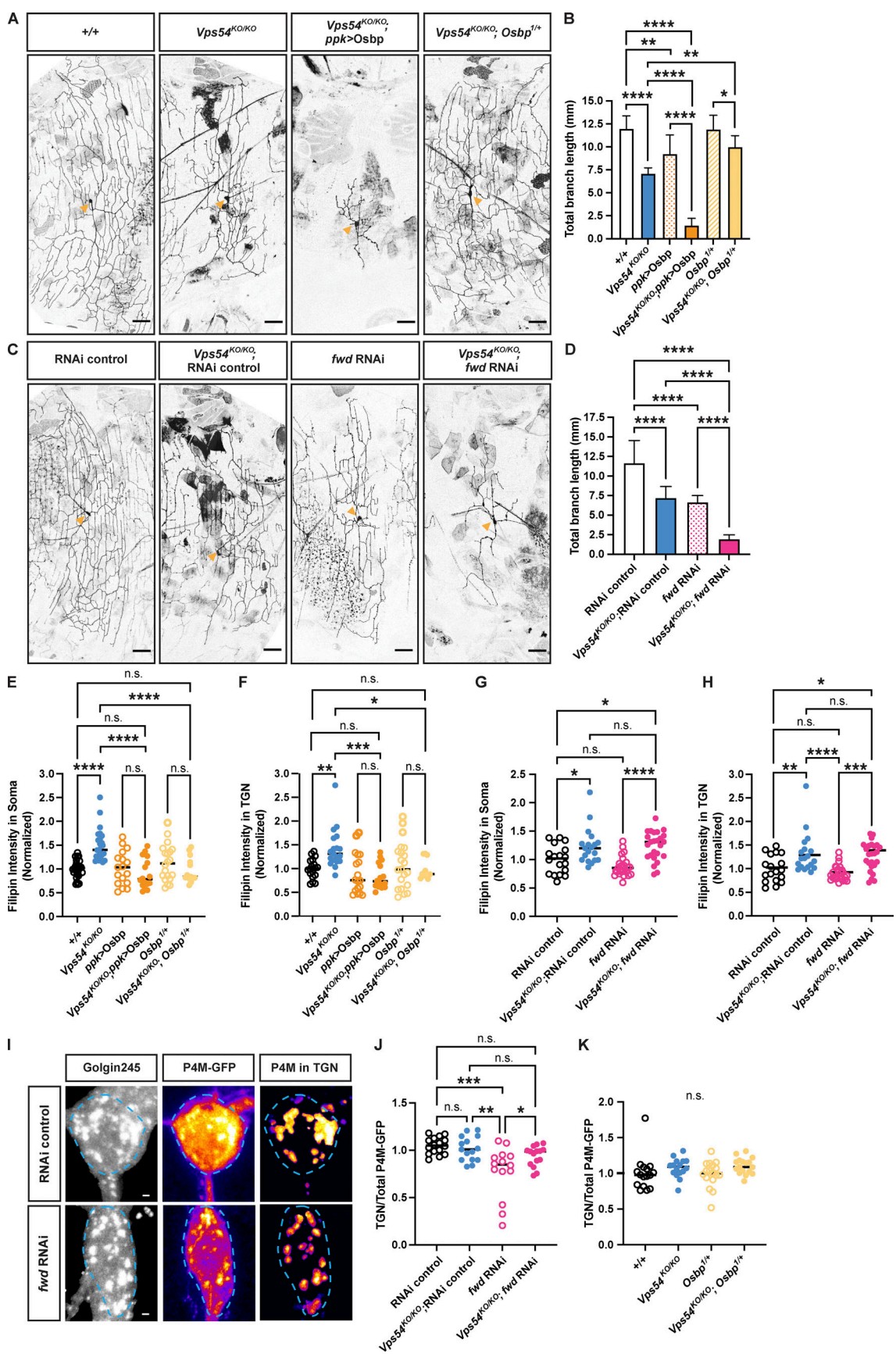

Figure 7. **Targeting specific lipid regulators at the TGN modulates GARP KO phenotypes. (A)** Representative maximum intensity z-projections of c4da neurons from 7-d-old males showing the effect of Osbp manipulation in the *Vps54^(KO/KO)* background. Scale bar = 50 μm. Yellow arrows point to the soma.

**(B)** Quantification of total dendrite branch length. Data presented as mean ± SD. $N \geq 10$ independent neurons/genotype except +/+ and $Vps54^{KO/KO}$ (both $n = 6$). Analyzed by one-way ANOVA with Tukey's post-test. ****$P < 0.0001$. +/+ vs. ppk > Osbp **$P = 0.0041$. $Vps54^{KO/KO}$ vs. $Vps54^{KO/KO}$; $Osbp^{1/+}$ **$P = 0.0027$. $Osbp^{1/+}$ vs. $Vps54^{KO/KO}$; $Osbp^{1/+}$ *$P = 0.039$. **(C)** Representative maximum intensity z-projections of c4da neurons from 7-d-old males showing the effect of *fwd* RNAi knockdown using the *ppk*-Gal4 to drive RNAi expression. **(D)** Quantification of total dendrite branch length. Data presented as mean ± SD. $N \geq 8$–$14$ independent neurons/genotype. ****$P < 0.0001$. **(E and F)** Quantification of filipin fluorescence intensity showing the effect of Osbp manipulation in $Vps54^{KO/KO}$ neurons at 96 h APF. $N \geq 13$ independent samples/genotype. Analyzed by one-way ANOVA with Šidák's post-test. **(E)** Total filipin staining in the soma. +/+ vs. $Vps54^{KO/KO}$, $Vps54^{KO/KO}$ vs. $Vps54^{KO/KO}$; ppk > Osbp, and $Vps54^{KO/KO}$ vs. $Vps54^{KO/KO}$; $Osbp^{1/+}$ ****$P < 0.0001$. All other comparisons n.s. $P > 0.76$. **(F)** TGN-associated filipin staining. +/+ vs. $Vps54^{KO/KO}$ **$P = 0.0079$. $Vps54^{KO/KO}$ vs. $Vps54^{KO/KO}$; ppk > Osbp ***$P = 0.0001$. $Vps54^{KO/KO}$ vs. $Vps54^{KO/KO}$; $Osbp^{1/+}$ *$P = 0.0362$. Other comparisons n.s. $P > 0.83$. **(G and H)** Quantification of filipin fluorescence intensity showing the effect of fwd knockdown in $Vps54^{KO/KO}$ neurons, $n \geq 18$ independent samples/genotype. **(G)** Total filipin staining in the soma. RNAi control vs. $Vps54^{KO/KO}$; RNAi control *$P = 0.0275$. RNAi control vs. $Vps54^{KO/KO}$; *fwd* RNAi *$P = 0.0123$. fwd RNAi vs. $Vps54^{KO/KO}$; *fwd* RNAi ****$P$, $0.0001$. N.s. $P > 0.38$. **(H)** TGN-associated filipin staining. RNAi control vs. $Vps54^{KO/KO}$; RNAi control **$P = 0.0099$. RNAi control vs. $Vps54^{KO/KO}$; *fwd* RNAi *$P = 0.0231$. fwd RNAi vs. $Vps54^{KO/KO}$; *fwd* RNAi ***$P = 0.0002$. N.s. $P > 0.69$. **(I)** Representative P4M-GFP images from RNAi control and *fwd* RNAi neurons. Left and middle panels show maximum intensity projections of Golgin245 and P4M-GFP, respectively. Right panel shows maximum intensity projection of extracted TGN-associated P4M-GFP signal. P4M-GFP images pseudocolored with the fire LUT. Dashed line shows soma area. Scale bar = 1 μm. **(J)** Quantification of TGN associated P4M-GFP fluorescence intensity from *fwd* RNAi samples. TGN P4M-GFP intensity levels normalized to total soma levels to account for variation in reporter expression. $N = 15$–$16$ independent samples/genotype. RNAi vs. *fwd* RNAi ***$P = 0.0001$. $Vps54^{KO/KO}$; *fwd* RNAi vs. $Vps54^{KO/KO}$; RNAi control **$P = 0.0011$. fwd RNAi vs. $Vps54^{KO/KO}$; *fwd* RNAi *$P = 0.0438$. N.S. $P > 0.23$. **(K)** Same as in J but for Osbp samples, n.s. $P > 0.22$.

transport to the TGN. Whether these contact sites exist in *Drosophila* is not yet clear as there is no obvious RELCH homolog. However, Rab11 has been shown to colocalize with TGN markers and to facilitate post-Golgi trafficking during photoreceptor development in flies (Satoh et al., 2005). Further, fwd can bind to Rab11, thereby localizing recycling endosomes with Golgi structures during cytokinesis (Polevoy et al., 2009), though it is not clear if this interaction permits sterol transfer. These studies suggest an intriguing hypothesis that an increase in TGN-Rab11⁺ recycling endosome contacts in $Vps54^{KO/KO}$ neurons may lead to sterol overloading, and further, that these interactions may disrupt post-Golgi secretory trafficking necessary for dendrite regrowth.

As sterol auxotrophs, *Drosophila* may utilize multiple pathways to transfer sterol from endolysosomes to the secretory pathway as they are unable to synthesize sterols endogenously. However, beyond the coordinated function of NPC1 and 2, other routes for sterol egress from endolysosomes remain poorly understood. Several recent studies in mammalian cells have focused on identifying mechanisms for sterol metabolism, and sterol transport specifically, in control cells and/or in cells in which NPC1 is either genetically or pharmacologically inhibited (Scott et al., 2015; van den Boomen et al., 2020; Trinh et al., 2020; Lu et al., 2022). We suggest that, given their unique reliance on dietary sterol, *Drosophila* is an ideal model in which to study mechanisms of sterol egress from the endolysosomal pathway. It will be important to conduct future studies to identify the mechanisms of sterol uptake and transport to the TGN in the absence of the GARP complex.

## Materials and methods

### *Drosophila* stocks
Flies were reared at 25°C in density-controlled vials containing standard cornmeal-molasses food. Vps50, Vps53, and Vps54 knockout lines were generated in this study as described below. Previously generated stocks include ppk-Gal4 (Grueber et al., 2002), ppk-Gal4, UAS-CD4-tdTomato or UAS-CD4-tdGFP lines (Han et al., 2011), Gal4²²¹, UAS-CD4-tdGFP (Grueber et al., 2003)

MARCM stock: yw SOP-FLP; FRT⁴⁰ᴬ tub Gal80; ppk-Gal4, UAS-CD4-tdGFP. The following fly lines were purchased from the Bloomington Stock Center: Chromosomal deficiencies deleting regions around the genes of interest: stock# 24372 (Vps50 Df: Df [2R]BSC348/CyO); stock# 7895 (Vps51 Df: Df[2R]Exel7158/CyO); stock# 27381 (Vps52 Df: Df[2L]BSC810/SM6a); stock# 23680 (Vps53 Df: Df[2L]BSC295); and stock# 7813 (Vps54 Df: Df[2L] Exel8022. RNAi lines: stock# 35787 (RNAi control UAS-mCherry in the VALIUM10 vector), stock# 50548 (Vps51 RNAi), stock# 27985 (Vps52 RNAi), stock# 38267 (Vps53 RNAi), stock# 38994 (Vps54 RNAi), stock# 35257 (fwd RNAi), stock# 27312 (Vap33 RNAi). Other lines: stock# 26693 (UAS-Vap-33-1), stock# 57348 ($Osbp^1$), stock# 57346 (UAS-Osbp), stock# 42716 (UAS-spinster-RFP), stock# 42714 (UAS-GFP-Lamp), stock# 64747 20XUAS-tdTomato-sec61β, stock# 1816 (FRT⁴⁰ᴬ), stock# 27893($Smid^{C161}$ Gal4). The following RNAi lines were purchased from the Vienna *Drosophila* Resource Center: stock #60200 (KK RNAi control), stock 108290 (Vps50 RNAi). We also generated the following recombinant lines: $Vps53^{KO}$, FRT⁴⁰ᴬ; $Vps54^{KO}$, FRT⁴⁰ᴬ; $Vps50^{KO}$, UAS-CD4-tdTomato; $Vps54^{KO}$, UAS-CD4-tdTomato; $Vps50^{KO}$, UAS-GFP-Lamp; $Vps54^{KO}$, UAS-GFP-Lamp; $Vps54^{KO}$, ppk-Gal4. The UAS-P4M-GFP line (Balakrishnan et al., 2018) was a generous gift from the Raghu lab.

### Molecular cloning
To generate the UAS-Vps50-3xHA line, Vps50 cDNA was amplified from DGRC clone FI23003. Restriction sites and a C-terminal 3xHA tag were added during amplification (primers listed in Table S1). The resulting amplification product was cloned into the Not and Kpn restriction sites in the pACU backbone. To generate the UAS-Vps53-3xHA line, Vps53 cDNA was amplified from DGRC clone clone FI1784. AttB sites were added during amplification. The amplification product was subsequently cloned in the pDONR-221 entry vector using BP-Clonase II (Invitrogen) and subsequently transferred to the pTWH vector (DGRC clone 1100) using LR-Clonase II (Invitrogen). The UAS-Scat line was generated using the expression ready pDNR-Dual-UAS-Scattered-Flag-HA plasmid (DGRC clone FMO06004).

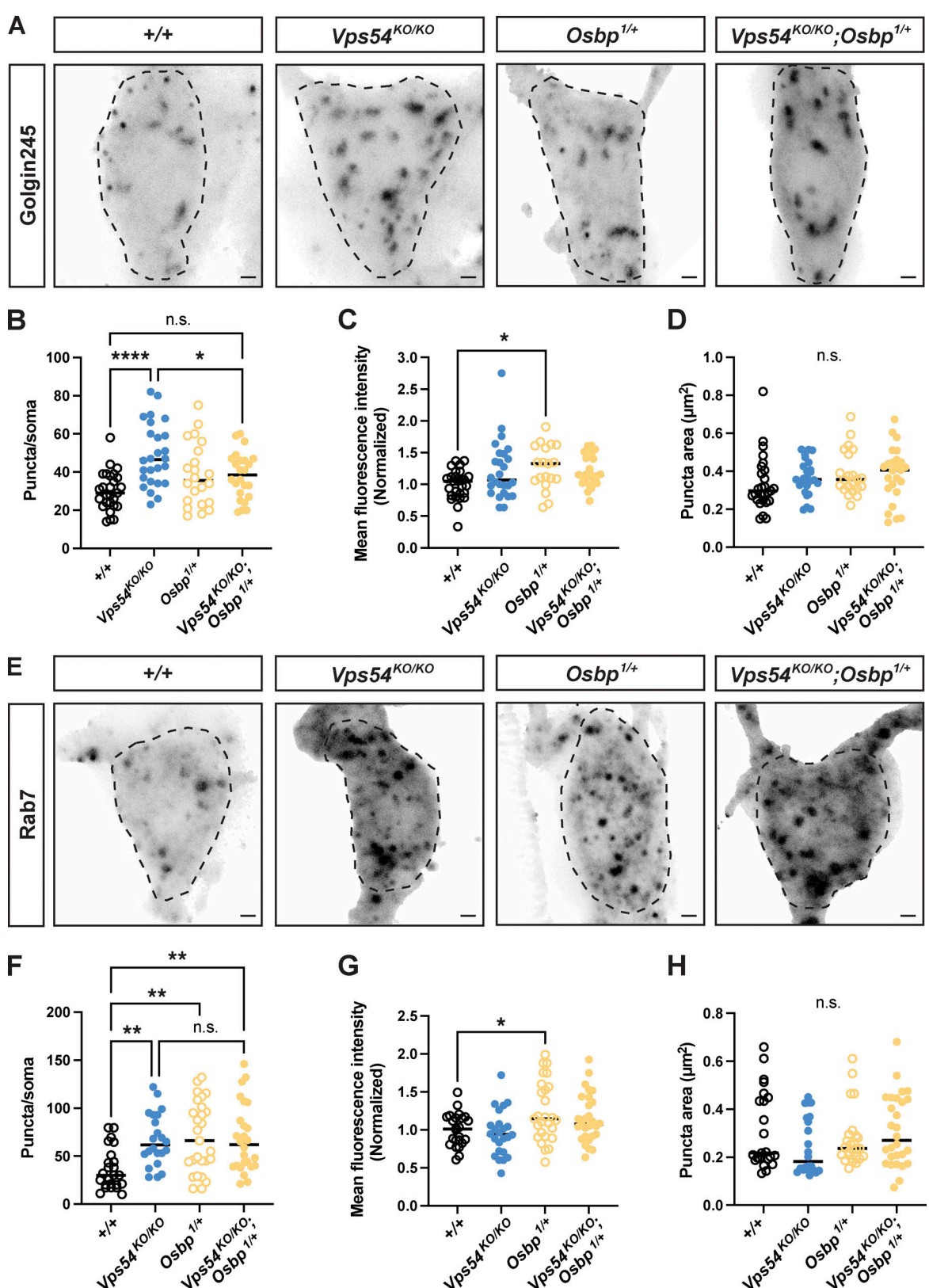

Figure 8. **Depletion of Osbp rescues TGN but not late endosomal morphology in GARP KO neurons. (A)** Representative maximum intensity z-projections of endogenous Golgin245 staining from c4da neurons in pupae 96 h APF. Dashed line shows soma area. Scale bar = 1 μm. For B–D, *n* = 22–27 independent samples/genotype. All data in this figure were analyzed by one-way ANOVA with Tukey's post-test. Samples were obtained from at least three independent experiments. **(B)** Quantification of the number of Golgin245 puncta/soma. *+/+* vs. *Vps54^{KO/KO}* ****P < 0.0001, *Vps54^{KO/KO}* vs. *Vps54^{KO/KO}; Osbp^{1/+}* *P = 0.0404. *+/+* vs. *Vps54^{KO/KO}; Osbp^{1/+}* n.s. P = 0.1728. **(C)** Golgin245 fluorescence intensity. *+/+* vs. *Osbp^{1/+}* *P = 0.0406. *+/+* vs. *Vps54^{KO/KO}* n.s. P = 0.0808, *+/+* vs. *Vps54^{KO/KO}; Osbp^{1/+}* P = 0.0683. **(D)** Golgin puncta area. n.s. P > 0.24. **(E)** Representative maximum intensity z-projections of endogenous Rab7 staining from

c4da neurons in pupae 96 h APF. **(F–H)** For F–H, *n* = 23–28 independent samples/genotype. **(F)** Quantification of the number of Rab7 puncta in the soma. *+/+* vs. *Vps54$^{KO/KO}$* **P = 0.0067. +/+ vs. *Osbp$^{1/+}$* **P = 0.0013. +/+ vs. *Vps54$^{KO/KO}$*; *Osbp$^{1/+}$* **P = 0.0029· Other comparisons n.s. P > 0.98. **(G)** Rab7 fluorescence intensity. +/+ vs. *Osbp$^{1/+}$* **P = 0.0404. Other comparisons P > 0.11. **(H)** Rab7 puncta area. All comparisons, n.s. P > 0.36.

The UAS-Vps50 plasmid was injected into the VK20 attP docking for φC-31–mediated integration (Bateman et al., 2006) site by Genetivision. The UAS-Vps53 and UAS-Scat plasmids were injected into Bloomington stock 24866 (M{vas-int.Dm}ZH-2A, PBac{y[+]-attP-9A}VK00019) by Rainbow Transgenics.

### Generation of knockout flies by CRISPR

Knockout flies were generated by CRISPR/CAS9 homology-dependent repair in which the gene of interest was replaced by an eye-specific dsRed cassette (see Fig. S1 for schematic). Guide RNA sequences were designed using the Fly CRISPR target finder (https://flycrispr.org/). Selected sequences were in the 5′UTR and -3′UTR of the gene of interest. Each guide RNA was cloned into the pU6-BbsI-chiRNA. Guide RNA sequences and genotyping primers can be found in Table S1.

The donor template was generated by cloning homology arms (∼1 kB upstream of the 5′ guide RNA sequence and ∼1 kB downstream of the 3′ guide RNA sequence; see Table S1 for primer sequences) into the pHD-dsRed-attB plasmid. For Vps50 and Vps53, 5′ homology arms were cloned into the NotI site, while 3′ homology arms were cloned into the SpeI site. For Vps54 (scat), the 5′ homology arm was cloned into the AarI site, while 3′ homology arm was cloned into the SapI site.

Guide RNA plasmids and donor plasmids were injected into isogenized *vasa*-cas9 flies (Rainbow Transgenics). DsRed+ flies were selected and crossed to balancers. DNA isolated from homozygous DsRed+ flies was used for initial genotyping. To generate a control line, the isogenized vasa-cas9 flies were treated in the same manner as DsRed+ flies. The resultant line was used as the control (+/+) unless otherwise indicated.

After initial screening, the DsRed cassette was removed by crossing to flies expressing Cre recombinase (specific line). DsRed-offspring from this cross were mated to a second chromosome balancer line. Homozygous progeny were used for genotyping to confirm the absence of the gene of interest (Fig. S1).

### RT-PCR

Total RNA was isolated from wandering third instar larvae by TRIzol/chloroform extraction and treated with the TURBO DNA-free reagent (Thermo Fisher Scientific) to remove genomic DNA. The High-Capacity cDNA Reverse Transcription Kit (Applied Biosystems) was used for cDNA synthesis. Primer sequences for *Vps50*, *Vps53*, *Vps54* and three internal controls can be found in Table S1. PCR reactions were performed on equal amounts of cDNA with SYBR Green PCR Master Mix (Applied Biosystems). PCR products were run on 1% agarose gels with the GeneRuler 1 kb Plus DNA Ladder (Thermo Fisher Scientific).

### Lifespan analysis

Lifespan analysis was conducted at 25°C. Groups of 10 age-matched flies were collected as they eclosed and transferred to yeasted vials containing standard cornmeal-molasses food. Every 3–4 d,

surviving flies were transferred to fresh vials and the number of dead and surviving flies were recorded. Flies were excluded from the study if they escaped, were accidentally crushed, or were stuck in food while still alive. Kaplan–Meier curves were generated in Prism (GraphPad) and analyzed by Mantel–Cox log rank test with Bonferroni correction for multiple comparisons.

### Antibodies

The following antibodies were used in this study: Mouse anti-*Drosophila* Rab7 hybridoma supernatant (1:5) and goat anti-*Drosophila* golgin245 (both developed and validated by S. Munroe [Riedel et al., 2016] and obtained from the Developmental Studies Hybridoma Bank). Rabbit anti-*Drosophila* Rab5 (1: 250, Ab31261; Abcam). Rat anti-tdTomato (1:500, EST203; Kerafast). Mouse anti-GFP (1:500, 1181446000; Sigma-Aldrich). Mouse anti-insect cathepsin L antibody (1:500, MAB22591; R&D Systems). Mouse anti-tubulin antibody (1:2,000, T9026; Sigma-Aldrich). The guinea pig anti-*Drosophila* scattered (Vps54) antibody was generated by and obtained from R. Sinka (Fári et al., 2016; 1: 400). Secondary antibodies for immunohistochemistry were anti-mouse, goat, rabbit, or rat labeled by Alexa 488, 555, or 647 (1:1,000; Thermo Fisher Scientific). Secondary antibodies for Western blot were anti-mouse HRP (1:1,000, 115-035-146; Jackson), anti-guinea pig HRP (1:500, A5545; Sigma-Aldrich), or anti-mouse IR-Dye 680 LT (1:20,000; LI-COR).

### Western blotting

10–20 whole larvae or 30–40 heads from 1-d-old flies were homogenized in 50 mM Tris-HCl, pH 7.4, 150 mM NaCl, 1% Triton X-100, 5 mM EDTA, 1 mM PMSF, and 1× complete protease inhibitor (Roche) using a pestle. Samples were then centrifuged for 10 min at 12,000 × *g*. Samples for gel electrophoresis were prepared 2× Laemmli Buffer (1610737; Bio-Rad) with 5% β-mercaptoethanol. Lysates were heated at 95°C for 10 min, followed by pulse centrifugation. Samples were loaded on 4–12% Bolt Bis-Tris Plus (Thermo Fisher Scientific) and run in NuPage MES buffer (Thermo Fisher Scientific). Proteins were then transferred to Immobilon-FL PVDF membrane (Millipore), blocked in 5% milk in TBST (Tris-buffered saline + 0.1% Tween-20). Primary antibodies were diluted in blocking solution and incubated overnight at 4°C. After washing with TBST, membranes were incubated with secondary antibodies for 2 h at room temperature. Membranes were then again washed before detection. Vps54/scat protein was detected using HRP secondary antibodies with the SuperSignal West Pico ECL chemiluminescent substrate (34580; Thermo Fisher Scientific) and scanned on a C-DiGit blot scanner. Cathepsin L Western blots were detected using LI-COR secondaries and scanned on the LI-COR Odyssey CLx.

### Imaging dendrite morphology

For c4da larval and pupal imaging, staged embryo collections were performed on yeasted grape agar plates at 25°C. Third

instar larvae (96 h AEL) were anesthetized in ether and whole mounted in glycerol. Staged pupae (72 or 96 h APF) were dissected from the pupae case and mounted on a custom acrylic disc (de Vault et al., 2018) without any anesthesia. For adult imaging, flies that eclosed within an 8-h time window were collected as age-matched adults. Flies were aged at 25°C in yeasted vials and were transferred to fresh vials every 3–4 d. Flies were anesthetized with $CO_2$ and whole mounted in glycerol. Z-stacks of dendrites of single neurons from independent larvae/pupae/flies were collected with a 0.5 µm z-step on a Leica SP5 laser-scanning confocal microscope equipped with a 20 × 0.75 N.A. oil immersion objective, HyD detectors on standard mode, and LAS X acquisition software. C4da neurons were visualized using the ppk-Gal4, UAS-CD4-tdTomato, or UAS-CD4-tdGFP lines (Han et al., 2011). $Vps53^{KO}$ MARCM clones were generated using the yw SOP-FLP; $FRT^{40A}$ tub Gal80; ppk-Gal4, UAS-CD4-tdGFP stock. C4da ddaC (in larvae) and v'ada (in adults) neurons were imaged. Larval class 1 ddaE neurons were visualized using Gal4$^{221}$, UAS-CD4-tdGFP (Grueber et al., 2003). Larvae were collected and mounted as above. To image adult class I ddaE neurons, UAS-CD4-tdTomato; $Vps50^{KO}$, UAS-CD4-tdTomato recombinant; or $Vps54^{KO}$, UAS-CD4-tdTomato recombinant flies were crossed to control or knockout lines containing the $Smid^{CI61}$ Gal4 (Shimono et al., 2009). Newly eclosed flies were collected and dissected (0–8 h after eclosion). Fillets were fixed with 4% paraformaldehyde for 20 min at room temperature. After washing, samples were incubated with anti-tdTomato antibody overnight at 4°C. The following day, samples were incubated with secondary antibody for 2 h at room temperature. Samples were then mounted with Diamond ProLong Anti-fade mounting reagent. Z-stacks of dendrites were collected with a 0.3 µm z-step on a Leica SP8 laser-scanning inverted confocal microscope equipped with a 40 × 1.3 N.A. oil immersion objective, HyD detectors on standard mode, and LAS X acquisition software. Single neurons from independent samples were used for image analysis.

### Imaging axonal morphology
$FRT^{40A}$, $Vps53^{KO}$ $FRT^{40A}$, and $Vps54^{KO}$ $FRT^{40A}$ were crossed to yw SOP-FLP; $FRT^{40A}$ tub Gal80; ppk-Gal4, UAS-CD4-tdGFP. After eclosion, adult flies were mounted on an acrylic disk, anesthetized with $CO_2$, and imaged on the Leica SP5 to screen for single c4da neuron clones in the abdomen. Flies with single clones were then aged at 25°C in yeasted vials and were transferred to fresh vials every 3–4 d. At 6–8 d after eclosion, the VNC was dissected in cold PBS and fixed in 4% paraformaldehyde for 20 min at room temperature. After washing and permeabilization with 0.5% Triton, VNC were blocked with 10% serum and then incubated with anti-GFP for 2 h at room temperature. After washing, samples were incubated with secondary antibody for 2 h at room temperature. VNC were mounted in Diamond ProLong Anti-fade mounting reagent. Z-stacks were collected with a 0.3 µm z-step on a Leica SP8 laser-scanning inverted confocal microscope equipped with a 40 × 1.3 N.A. oil immersion objective, 1.2× zoom digital zoom, HyD detectors on standard mode, and LAS X acquisition software.

### Immunohistochemistry and filipin staining
Larvae, pupae, or adults were filleted and fixed in 4% paraformaldehyde for 20 min followed by permeabilization using 0.5%

Triton X. Fillet preps were blocked in 10% serum and then incubated with primary antibody while rotating overnight at 4°C. After primary antibody was washed away, fillets were incubated with secondary antibody for 2 h while rotating at room temperature. Fillets were mounted in Diamond ProLong Anti-fade mounting reagent and imaged on Z-stacks for analysis of organelles or filipin staining were collected with a 0.25 µm z-step on a Leica SP8 laser-scanning inverted confocal microscope equipped with a 63 × 1.4 N/A. oil immersion objective, 3–6× zoom digital zoom, HyD detectors on standard mode, and LAS X acquisition software. Samples to be stained with filipin were first fixed and then stained with 5 µg/ml filipin in PBS for 2 h at room temperature without permeabilization. If filipin-stained samples were also to be stained with antibodies, samples were then permeabilized and stained using the same procedure described above. Permeabilized samples were incubated with either anti-tdTomato or anti-GFP antibodies to boost the neuronal membrane marker signal. P4M-GFP samples were co-stained with anti-GFP and Golgin245. For all organelle experiments, 1–4 neurons per independent sample were imaged and the average values/independent sample were used to generate graphs.

### Image analysis
The experimenter was blinded to genotype during all image processing and analysis. Image analysis for dendrite and organelle morphology was performed in ImageJ Fiji (http://fiji.sc). Morphological analysis of dendrite arbors was performed on maximum projections of z-stacks. Dendrite arbors were reconstructed using the Simple Neurite Tracer (Longair et al., 2011). Total dendrite branch length is the summed length of all dendrite branches from a single neuron reconstruction. Sholl Analysis was performed using the built-in Sholl Analysis function. To obtain coverage index, neuron area was measured and divided by hemisegment area as in Parrish et al. (2000).

For organelle analysis, masks were generated from z-stacks of the tdTomato or tdGFP neuronal membrane marker and applied to z-stacks of organelle staining to isolate organelles in neurons from background (neuronal organelle image). Maximum projections of organelle staining were further processed as 8-bit binary images by applying the remove outliers, fill holes, dilate, and watershed functions. The Analyze Particles function was used to create ROI around the organelles and ROI then transferred to neuronal organelle maximum intensity projection, and used measure puncta number, area, and mean fluorescence intensity. To measure filipin levels in organelles, a mask was created on the organelle marker z-stack and then applied to z-stacks of filipin staining. Filipin intensity levels were then measured on maximum projections of the masked images.

Image analysis of axon terminals was performed in Imaris 5.5 software. Axon terminal branches were manually traced in 3D view using the filament function. Total branch length is the summed length of all axon terminal branches in the abdominal neuromere of the ventral nerve cord.

### Statistical analysis
Statistical analyses were performed in GraphPad Prism software. Survival curves were analyzed by Log-Rank Mantel-Cox

test with Bonferroni multiple comparisons correction. Comparison of two genotypes was performed by unpaired two-sided *t* test for data sets with a normal distribution. The non-parametric unpaired two-sided Mann–Whitney U-test was performed for data sets that were not normally distributed. Data distribution was tested for normality using a D'Agostino and Pearson omnibus K2 test and a $P > 0.05$ was used to determine normality. Comparison of three or more genotypes was performed with one-way ANOVA with either Tukey's or Šidák's multiple comparison's test. Analysis of two or more genotypes over time was carried out by two-way ANOVA with either Tukey's or Šidák's multiple comparison's test.

### Online supplemental material

Fig. S1 supports Fig. 1 and describes the CRISPR strategy and initial molecular characterization of the knock-out lines. Fig. S2 supports Fig. 3 and presents results from RNAi knockdown of GARP and EARP components in adult c4da neurons, quantification of larval c4da dendritic arbors, additional analyses related to Fig. 3 E, and quantification of adult c4da axon terminals of MARCM clones. Fig. S3 supports Fig. 5 and presents additional parameters for organelle data in the soma and proximal dendrites from adults and larvae, as well as Western blot data showing cathepsin L processing. Fig. S4 supports Fig. 6 and shows images and quantification of filipin in lysosomes and the ER. Fig. S5 supports Fig. 7 and presents results from over-expression or RNAi knockdown of Vap33 in control and *Vps54$^{KO/KO}$* neurons. Table S1 lists the primers used for generating and genotyping CRISPR knockouts. Table S2 lists P values for all comparisons for Fig. 3 E.

### Data availability

All data, fly stocks, and other reagents generated in this study are available to the scientific community upon request.

### Acknowledgments

We thank Maja Petkovic and Kai Li for critical reading of the manuscript and members of the Jan lab for discussion. We thank Rita Sinka for providing the scattered (Vps54) antibody and Padinjat Raghu for the UAS-P4M-GFP PI4P reporter flies.

This work was supported by an NIH National Institute of Neurological Disorders and Stroke grant (R35NS097227) to Y.N. Jan. Both L.Y. Jan and Y.N. Jan are investigators of the Howard Hughes Medical Institute.

The authors declare no competing financial interests.

Author contributions: Conceptualization: C. O'Brien and Y.N. Jan. Resources: C. O'Brien and S.H. Younger. Investigation and formal analysis: C. O'Brien. Writing—original draft: C. O'Brien. Writing—review and editing: C. O'Brien, S.H. Younger, L.Y. Jan, Y.N. Jan. Funding Acquisition: L.Y. Jan and Y.N. Jan.

Submitted: 3 January 2022

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

# Supplemental material

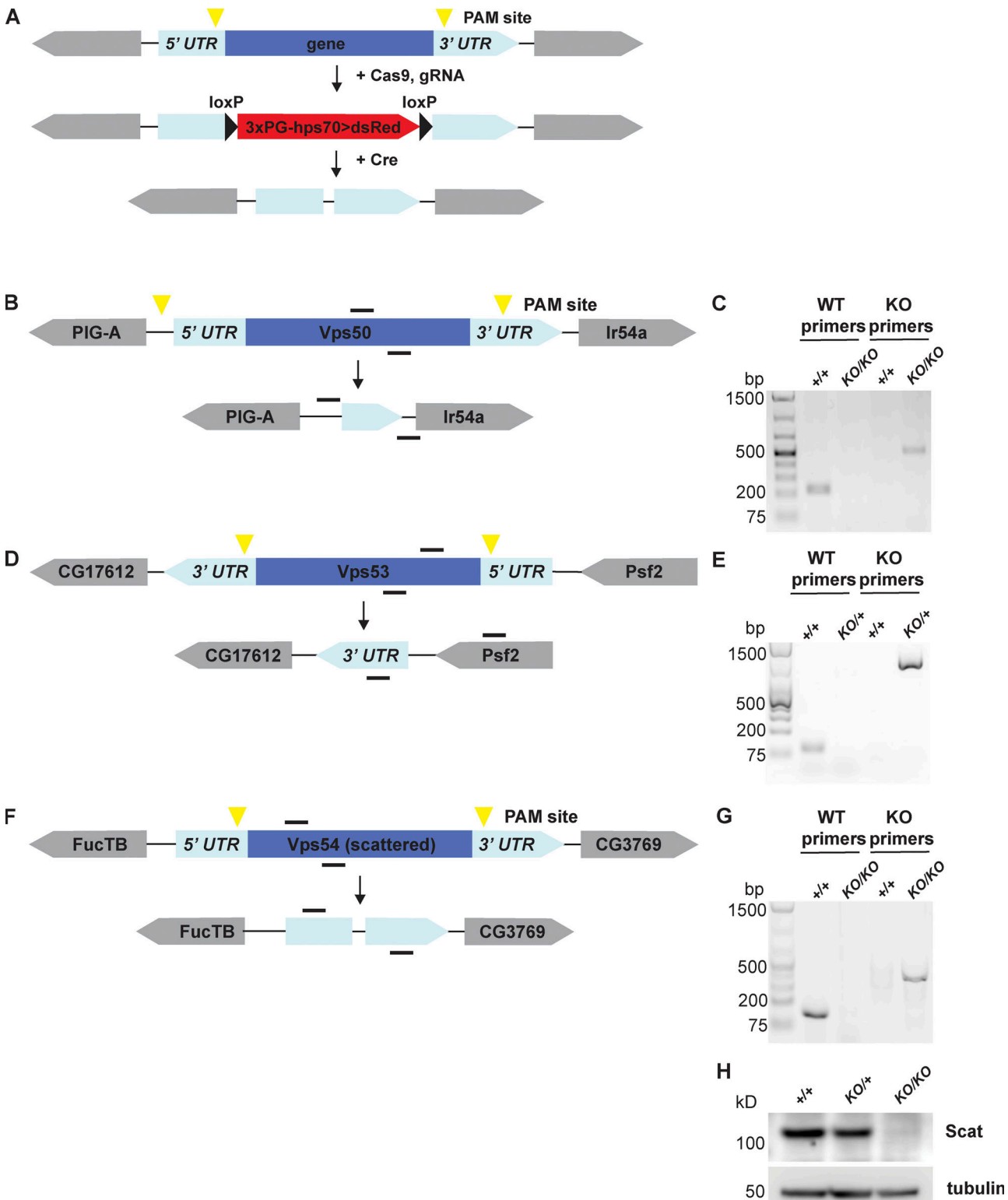

**Figure S1.** **Generation of GARP and EARP knockout flies. (A)** General schematic of CRISPR knockout strategy. Guide RNAs recognize sequences around PAM sites (yellow triangles). DsRed cassette flanked by loxP sites was knocked-in in place of the gene of interest to generate DsRed+ knockout (KO) lines. DsRed cassette was removed by crossing to a Cre recombinase line to generate the final knockouts. **(B, D, and F)** Schematic of Vps50, Vps53, and scattered (Vps54) wild-type genes and knockouts, respectively. Genes are shown in their relative orientation in the genome. Black lines above indicate hybridization sites for genotyping primers. **(C, E, and G)** Agarose gel of genotyping PCR for Vps50, Vps53, and Vps54 knockout lines, respectively. +/+ = w[1118]. For Vps50 and Vps54, DNA was isolated from adult males. Because Vps53[KO] is lethal in the pupal stage, DNA was isolated from wandering third instar larvae. Bp = base pairs. **(H)** Western blot of head lysates from control, Vps54[KO/+] and Vps54[KO/KO] larvae probed with antibodies raised against Vps54/scattered and tubulin (loading control). See Table S1 for cloning and genotyping primer sequences. Source data are available for this figure: SourceData FS1.

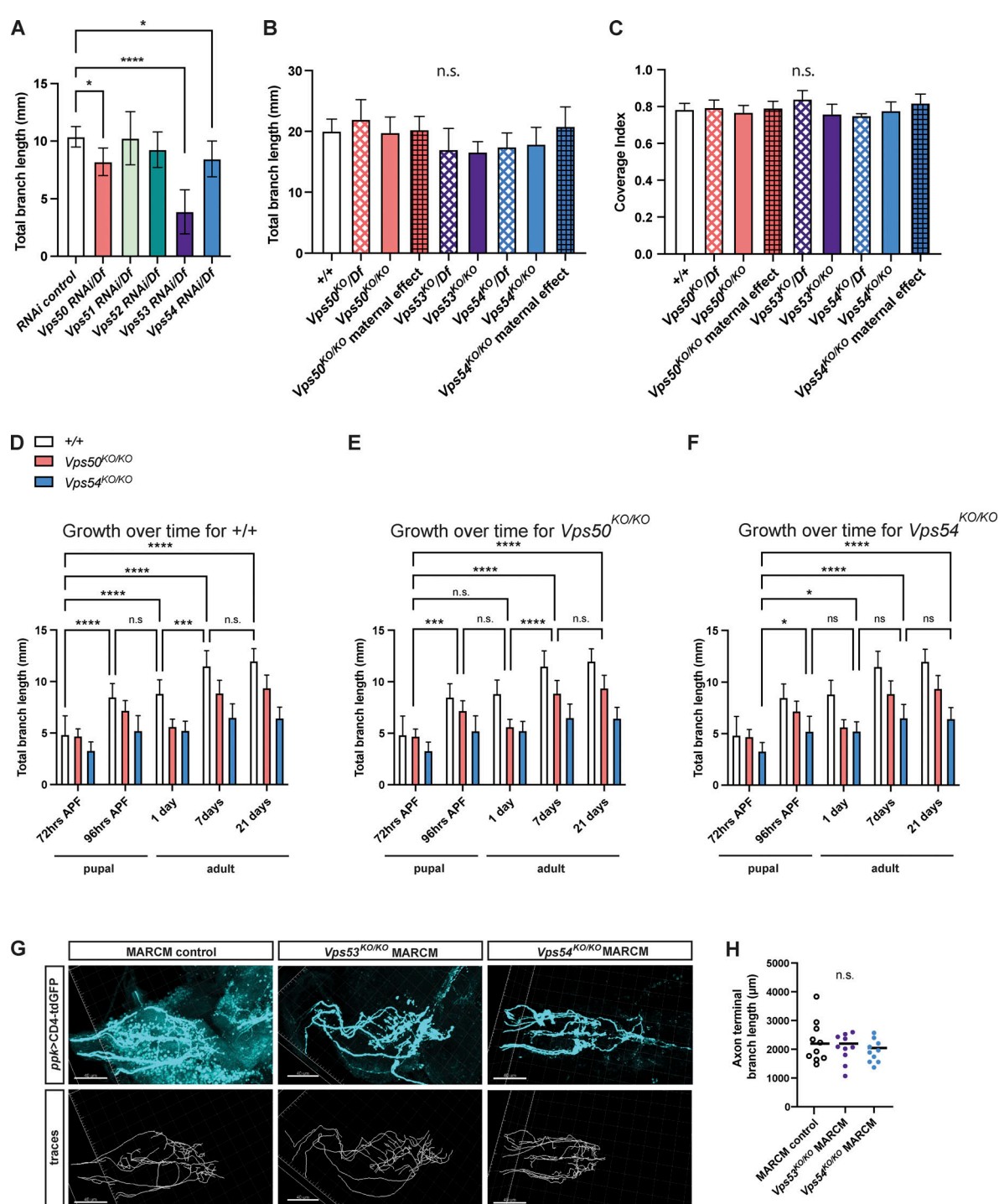

Figure S2. **Additional support for developmental emergence of a dendritic, but not axonal, phenotype. (A)** Quantification of total dendrite branch length of c4da neurons expressing RNAi against GARP and EARP complex components in 7 d adult males. RNAi are driven by ppk-Gal4 in a heterozygous chromosomal deficiency background, n = 7–13 independent neurons/genotype. RNAi control vs. Vps50 RNAi/Df *P = 0.0217, RNAi control vs. Vps54 RNAi/Df *P = 0.0238, RNAi control vs. Vps53 RNAi/Df ****P < 0.0001, remaining comparisons, P > 0.31. **(B)** Total dendrite branch length of c4da neurons from 3rd instar larvae 96 h after egg lay (AEL). For maternal effect samples, we examined homozygous KO flies from homozygous KO females crossed to heterozygous KO males. Comparison of each genotype to +/+ P > 0.16. **(C)** Coverage index (neuron area/receptive field area) of neurons from 96 h AEL larvae. Comparison of each genotype to +/+, P > 0.11. For B and C, n = 6–9 neurons/genotype. A–C were analyzed by one-way ANOVA with Tukey's post-test. Data presented as mean ± SD. **(D–F)** Additional analysis for data in Fig. 3 E. Graphs show statistics comparing total dendrite length across timepoints for (D) +/+, (E) Vps50$^{KO/KO}$, and (F) Vps54$^{KO/KO}$. Analyzed by two-way ANOVA with Tukey's post-test. Data presented as mean ± SD. Please see Table S2 for the list of P-values. **(G and H)** Representative images of axon terminals of FRT$^{40A}$ MARCM control, Vps53$^{KO/KO}$ MARCM, and Vps54$^{KO/KO}$ MARCM clones from 6- to 8-d-old adult ventral nerve cord. Top, CD4-tdGFP labeling the neuronal membrane. Bottom: traces of axon terminals. Scale bar = 40 μm. **(H)** Quantification of axon terminal branch length, n = 10 independent clones/genotype. Analyzed by one-way ANOVA with Tukey's post-test, n.s. P > 0.41. Samples for all experiments were collected from at least three independent experiments.

Figure S3. **Additional analyses of organelle phenotypes in** Fig. 5. **(A–E)** Additional quantification of Rab5 data from Fig. 5, A and B. Quantification of (A) puncta area and (B) mean fluorescence intensity in the soma. $n$ = 16–18 independent samples/genotype. For all graphs in this figure, each data point represents the average of 1–3 cells/sample. Data for each organelle marker was obtained from at least three independent experiments. All organelle data were analyzed by one-way ANOVA with Tukey's post-test. All comparisons n.s. P > 0.10. Quantification of Rab5 (C) puncta number/10 μm of dendrite length, (D) puncta area, and (E) mean fluorescence intensity in proximal dendrites. All comparisons were n.s. P > 0.15. **(F–J)** Additional quantification of Rab7 data from Fig. 5, C and D, $n$ = 17–20 independent samples/genotype. Quantification of (F) Rab7 puncta area. +/+ vs. $Vps50^{KO/KO}$ n.s. P = 0.0526. All other comparisons n.s. P > 0.14. **(G)** Rab7 mean fluorescence intensity in the soma. All comparisons n.s. P > 0.14. **(H–J)** Quantification of Rab7 in proximal dendrites. **(H)** Puncta number/10 μm of dendrite length, n.s. P > 0.34 for all comparisons. **(I)** Rab7 puncta area. +/+ vs. $Vps50^{KO/KO}$ puncta area n.s. P = 0.077. Other comparisons n.s. P > 0.25. **(J)** Rab7 mean fluorescence intensity, n.s. P > 0.25. **(K and L)** Additional quantification of GFP-Lamp data from Fig. 5, E and F. $N$ = 23 independent samples/ genotype. **(K)** GFP-Lamp puncta area +/+ vs. $Vps54^{KO/KO}$; ppk > Vps54 n.s. P = 0.0581. All other comparisons n.s. P > 0.44. **(L)** GFP-Lamp mean fluorescence intensity. $Vps54^{KO/KO}$ vs. $Vps54^{KO/KO}$; ppk > Vps54 **P = 0.0045. All other comparisons n.s. P > 0.10. **(M–O)** Quantification of spin-RFP puncta in the soma. $N$ = 10–12 independent samples/genotype. **(M)** Spin-RFP puncta number/soma. All comparisons n.s. P > 0.07. **(N)** Spin-RFP puncta area +/+ vs. $Vps54^{KO/KO}$ ***P = 0.0003. +/+ vs. $Vps50^{KO/KO}$ n.s. P = 0.9942. **(O)** Quantification of spin-RFP mean fluorescence intensity. All comparisons n.s. P > 0.09. **(P and Q)** Quantification of endosomes from the soma of class IV da neurons from larvae 96 h after egg lay for (P) rab5 ($n$ = 11–15 independent samples/genotype) and (Q) rab7 ($n$ = 10–14 independent samples/genotype). **(R)** Representative Western blot of head lysates from 1 d old +/+, $Vps50^{KO/KO}$ and $Vps54^{KO/KO}$ flies probed with antibodies against cathepsin L and tubulin. **(S and T)** Quantification of the (S) immature and (T) mature forms of cathepsin L. Samples are technical triplicates from two independent experiments. Analyzed by one-way ANOVA with Tukey's post-test, n.s. P > 0.23. U-Y Additional quantification of Golgin245 data from Fig. 5, G and H. $N$ = 18–21 independent samples/genotype. **(U and V)** Quantification of Golgin245 (U) puncta area and (V) mean fluorescence intensity from the soma. All comparisons n.s. P > 0.5. **(W–Y)** Quantification of Golgin245 (W) puncta number/10 μm of dendrite length, (X) puncta area and (Y) mean fluorescence intensity in proximal dendrites. All comparisons n.s. P > 0.14. Source data are available for this figure: SourceData FS3.

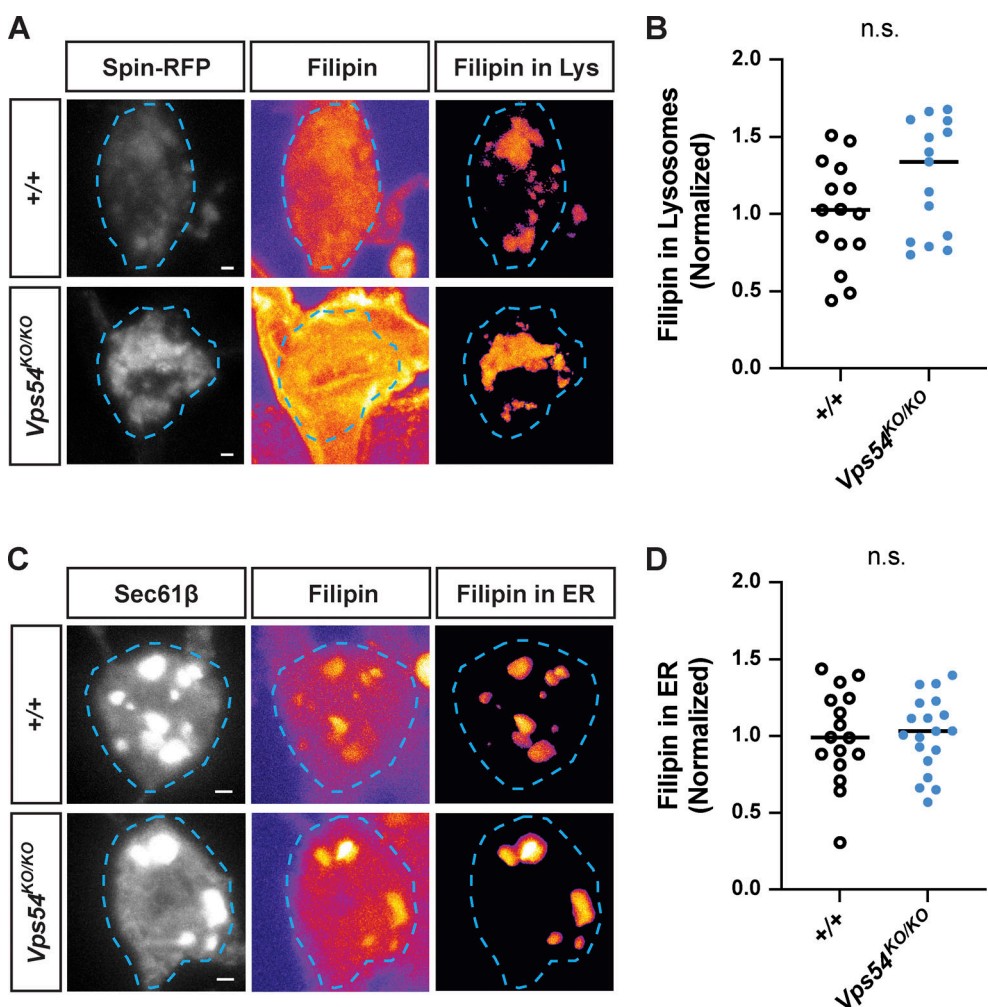

Figure S4. **Sterol accumulation in GARP deficient neurons. (A)** Filipin staining in lysosomes. For each genotype, left and middle panels show maximum intensity projections of Spin-RFP and filipin staining from 96 h APF pupae. Filipin images are pesudocolored with the fire LUT which cooler colors indicate lower and hotter colors indicate higher fluorescence intensity values. Scale bar = 1 μm. To obtain the filipin signal from lysosomes, a mask was generated using the Spin-RFP z-stack and applied to the filipin z-stack. Right panel shows maximum intensity projection of the extracted filipin signal in Spin+-lysosomes. **(B)** Quantification of lysosome-associated filipin staining. Analyzed by two-sided Mann-Whitney U test, n.s. P = 0.1064. N = 15 independent samples/genotype. **(C)** As in A, except with Sec61β to obtain filipin signal in the ER. **(D)** Quantification of ER-associated filipin signal. Analyzed by unpaired two-sided *t* test, n.s. P = 0.9006. N = 16–19 independent samples/genotype.

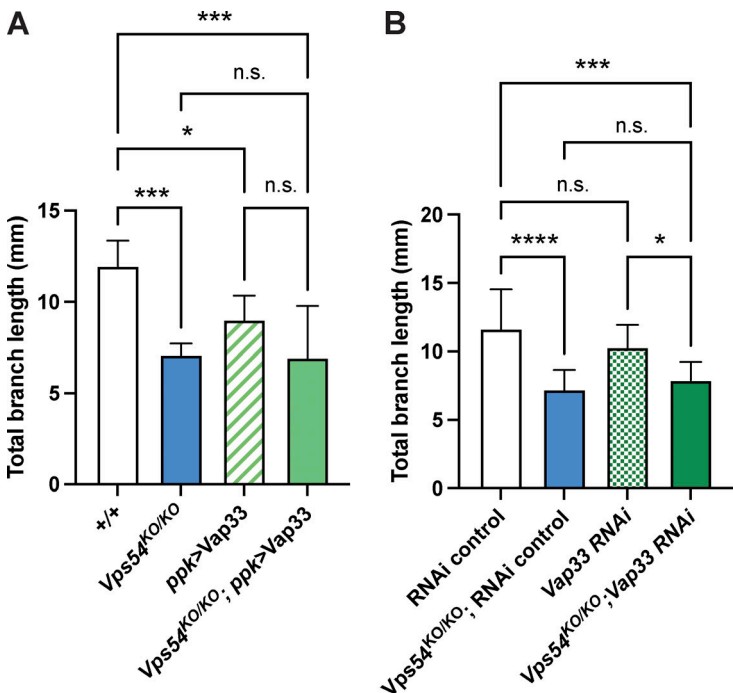

Figure S5. **Changing Vap33 expression does not affect dendrites in *Vps54^{KO/KO}* neurons** Quantification of total dendrite branch length of class IV da neurons in 7-d-old adults. Analyzed by one-way ANOVA with Tukey's post-test. Data presented as mean ± SD. **(A)** Effect of overexpression of Vap33 in c4da neurons. *+/+* vs. *Vps54^{KO/KO}* \*\*\*P = 0.0007. *+/+* vs. *ppk* > Vap33 \*P = 0.0265. *+/+* vs. *Vps54^{KO/KO}*; *ppk* > Vap33 \*\*\*P = 0.0001. *Vps54^{KO/KO}* vs. *Vps54^{KO/KO}*; *ppk* > Vap33 n.s. P = 0.9984. **(B)** Effect of Vap33 knockdown in c4da neurons. *RNAi* control vs. *Vps54^{KO/KO}*; *RNAi* control \*\*\*\*P < 0.0001. *RNAi* control vs. *Vps54^{KO/KO}*; *Vap33* RNAi \*\*\*P = 0.0009. *Vps54^{KO/KO}*; *RNAi* control vs. *Vps54^{KO/KO}*; *Vap33* RNAi n.s. P = 0.8367. *N* = 8–10 independent neurons for all genotypes except *+/+* and *Vps54^{KO/KO}* in A (*n* = 6 neurons for both).

**Provided online are Table S1 and Table S2. Table S1 lists primers used for generating and genotyping knockout flies by CRISPR. Table S2 shows additional statistics for Fig. 3 E.**

