## [Peer Review File · The Journal of Cell Biology]

The GARP complex prevents sterol accumulation at the trans-Golgi network during dendrite remodeling

Caitlin O'Brien, Susan Younger, Lily Jan, and Yuh-Nung Jan

Corresponding Author(s): Yuh-Nung Jan, Howard Hughes Medical Institute

Review Timeline:

Submission Date:	2022-01-03
Editorial Decision:	2022-01-11
Revision Received:	2022-08-11
Editorial Decision:	2022-09-14
Revision Received:	2022-09-18

Monitoring Editor: Marc Freeman

Scientific Editor: Lucia Morgado-Palacin

Transaction Report:

DOI: <https://doi.org/10.1083/jcb.202112108>

Revision 0

Review #1

1. How much time do you estimate the authors will need to complete the suggested revisions:

Estimated time to Complete Revisions (Required)

(Decision Recommendation)

Between 1 and 3 months

2. Evidence, reproducibility and clarity:

Evidence, reproducibility and clarity (Required)

****Summary:****

O'Brien and colleagues use *Drosophila* dendrite development to dissect the roles of the GARP and EARP vesicular trafficking complexes in the development of neuronal morphology. By making complex-specific KOs they investigate the role of each complex in the growth, pruning and re-growth of sensory dendrites and conclude that the GARP, but not EARP, complex is required for proper dendrite development by limiting sterol accumulation in the neuronal TGN.

****Major comments:****

While the data presented clearly support a role for GARP in regulating sterol levels to support dendritic growth, they do not inter current for suffice to exclude a role for EARP as important analyses to allow such a clear cut conclusion are either insufficient or missing. If the authors wish to maintain this claim - as suggested by the title of the manuscript - further analyses are essential.

1- Figure 3E shows that whereas both Vps50 and Vps54 mutations reduce dendritic complexity, the Vps54 phenotype appears earlier (96h APF). Furthermore, at 7 days dendrites appear to grow again but at a slower rate than controls. This begs the question of whether these mutations are causing a delay rather than a block in the regrowth after pruning and whether the growth will eventually be normal a few days later or whether it will stop at some point.

2- In Figure 4, the two mutations appear to have statistically differential effects on Rab5 and Rab7 puncta even though the data mean and distribution seem very similar. Interestingly, in each case the non-significant effect is associated with a smaller sample size. Given that the overall

sample sizes used are rather small for such highly variable data, this could easily cause a statistical anomaly due to sampling bias. The sample size should be made uniform across all genotypes and should ideally be at least doubled.

3- Perhaps the most important issue related to Figure 6 where the authors find that there is no sterol accumulation at 96h APF in the Vps50 mutant. However, even that the dendritic phenotype is slower to appear in this mutant compared to the Vps54, are the authors sure that the accumulation is not just slower? This should be examined using the same temporal sequence used for Vps54 shown in Figure 6C. In addition, the fact that sterol accumulation returns to normal in the Vps54 mutant at 1 day, supports the notion of a delay phenotype (see point 1 above).

These issues should be experimentally addressed to see if the data fully support the initial conclusions, or if the conclusions should be modified to suggest differential contribution of the two complexes to the process being studied and to a developmental delay phenotype.

3. Significance:

Significance (Required)

The study advances our understanding of the role of regulation of lipid storage in sculpting neuronal morphology during development.

Review #2

1. How much time do you estimate the authors will need to complete the suggested revisions:

Estimated time to Complete Revisions (Required)

(Decision Recommendation)

Between 1 and 3 months

2. Evidence, reproducibility and clarity:

Evidence, reproducibility and clarity (Required)

This manuscript presents a solid genetic analysis of components of the GARP and EARP complexes. The analysis is focused on a specialized type of sensory neurons i.e. class IV da neurons in Drosophila larvae. The authors show that loss of multiple components (VPS50-54) disrupt dendrite morphogenesis in c4da neurons in distinct ways. Additional genetic interaction

studies further support the notion of functional differences of GARP and EARP in vivo.

Overall this is a solid study and with one exception (see below) I have little concern regarding the presented experiments. I do, however, find the exclusive focus on a highly specialized cell type c4da somewhat problematic.

****Concerns:****

Experimental concern: It is stated that loss of VPS50 and VPS54 only causes dendrite morphogenesis defects. However, the corresponding supplemental figure S2c (which is not referenced in the text), is not suited to address this question. Axonal arborization, in particular terminal arbors, are not visible in samples where multiple/all c4da axons are labeled simultaneously (Fig. S2c). Analogous to the dendrite analysis of c4da neurons single cell resolution is essential to examine this in a meaningful way. Likely, however, c4da neurons may not be a good choice to address this question.

It would be important to know, whether the dendrite morphogenesis defect is indeed a developmental patterning defect or rather a "scaling" defect due to the fact that da neurons increase their size (but not necessarily their projection pattern) during larval maturation.

Overall, I am concerned whether the data shown here can be generalized. The cd4a neurons are rather extreme cell types due to their very large dendritic compartment. It seems quite possible that many other neurons may not have a comparable sensitivity to the supply of lipids/sterols. This type of question can only be addressed if other types of neurons/dendrites are examined. Are class 2 or class 3 da neurons showing any defects in VPS mutants?

3. Significance:

Significance (Required)

At this point i am not convinced that the findings can be generalized. The c4da neuron is really an extreme cell type with a massive disproportionate increase in membrane extensions. This is rather unusual and other neuron types should be tested.

Review #3

1. How much time do you estimate the authors will need to complete the suggested revisions:

Estimated time to Complete Revisions (Required)

(Decision Recommendation)

Less than 1 month

2. Evidence, reproducibility and clarity:

Evidence, reproducibility and clarity (Required)

O'Brien et al report how deficiency in GARP specific protein VPS54 or the EARP specific protein VPS50 affects the developmental dendritic remodeling of multidendritic class IV da (c4da) neurons in Drosophila. The main findings are that while both complexes play a role in dendritic remodeling, VPS54 deficiency leads to lipid accumulation in the trans-Golgi network (TGN). Manipulating sterols at the TGN affects dendritic remodeling suggesting that lipid accumulation is responsible for control of neuron morphology in this model. Overall, the data is interesting and the authors develop the experiments enough to be convincing on their major claims. However, a few aspects need clarification and perhaps revisiting conclusions.

Major comments

- The statistical analyses generally look appropriate but it would be critical to clarify what N means in every case. For example in figure 2 the authors state n=8 without clarifying if this is n=8 animals or n=8 neurons. N should always be the number of animals, but then the n of independent cells counted should also be indicated. Typically, one would either pre-average per genotype or use a mixed model that includes N of animals and n of cells (or similar).
- Please add details of how experiments were blinded to genotype
- Some of the experiments include multiple genotypes and so it would be important to show all in all figures. For example, figure 4B,D show four groups but figure 4F, presumably from the same set of animals, shows only three. Addition of the rescue genotype to 4F is particularly important here so should be shown. The same concern is valid for figure 5, where puncta number and area must be available.
- Related to figure 5, please provide validation of the staining of the TGN. Typically, one would expect trans Golgi to be close to the nucleus with at least some extended stacks. A Golgin245 knockout would be ideal.
- This concern is amplified by the images in figure 6 of the filipin staining, that are more obviously perinuclear. However, the two sets of images in 6A and 6D, where co-staining with Golgin245 is shown, look very different. Improved images are required and it may be helpful to use supplementary information to show additional examples of the staining.
- For figures 6F, G please show examples of staining for late endosomes and lysosome with appropriate validation.
- For the lipid regulation experiments in figure 7, please use an orthogonal approach to show that the Osbp and fwd RNAi had the expected effects on lipid accumulation.
- Figure 8 needs examples of the TGN and late endosome morphology.

Minor comments

- The title of figure 2 is inaccurate, at least if I understand the experiment, as it does not show neuron-specific knockout but instead whole body knockout with neuron rescue. Please rephrase.

- For ease of reading, it would be helpful to show genotypes in the same order in all figures (see 4B, 4D)

3. Significance:

Significance (Required)

The advance here is to nominate lipid accumulation at the trans Golgi network (TGN) is sufficient to affect dendritic remodeling during development. Although the work is performed in a model system, it may have relevance to human neurodevelopmental disorders caused by mutations in the orthologous genes. The work will be of highest relevance to developmental neurobiologists, particularly those working on GARP or EARP mutations and those who use *Drosophila* as an appropriate model for neurodevelopment.

Revision Plan

Manuscript number: RC-2021-01087

Corresponding author(s): Yuh Nung Jan

1. General Statements

We thank the reviewers for their helpful comments. We believe that we will be able to address all of their concerns and suggestions. We have highlighted our responses in the revision plan and the changes we have already made to the manuscript in blue text. For figures where we have added data or analyses at the request of reviewers, we have highlighted the corresponding text in the figure legends.

2. Description of the planned revisions

Reviewer #1

2- In Figure 4, the two mutations appear to have statistically differential effects on Rab5 and Rab7 puncta even though the data mean and distribution seem very similar. Interestingly, in each case the non-significant effect is associated with a smaller sample size. Given that the overall sample sizes used are rather small for such highly variable data, this could easily cause a statistical anomaly due to sampling bias. The sample size should be made uniform across all genotypes and should ideally be at least doubled.

We will repeat this staining to increase the n to at least double this number, and adjust our conclusions if need be, in the revised manuscript..

3- Perhaps the most important issue related to Figure 6 where the authors find that there is no sterol accumulation at 96h APF in the Vps50 mutant. However, even that the dendritic phenotype is slower to appear in this mutant compared to the Vps54, are the authors sure that the accumulation is not just slower? This should be examined using the same temporal sequence used for Vps54 shown in Future 6 C. In addition, the fact that sterol accumulation returns to normal in the Vps54 mutant at 1 day, supports the notion of a delay phenotype (see point 1 above).

These issues should be experimentally addressed to see if the data fully support the initial conclusions, or if the conclusions should be modified to suggest differential contribution of the two complexes to the process being studied and to a developmental delay phenotype.

Revision Plan

We had included the filipin staining for $Vps50^{KO/KO}$ at 1 day in Figure S4 A (which did not show a significant difference from control). We did not collect data for this genotype at 72hrs APF because the dendritic length phenotype didn't appear until later, and so we did not include $Vps50^{KO/KO}$ in the full time-course in Fig 6 C. We will collect additional data so that we can include $Vps50^{KO/KO}$ at all timepoints in this figure in the revised manuscript.

Reviewer #2

It is stated that loss of VPS50 and VPS54 only causes dendrite morphogenesis defects. However, the corresponding supplemental figure S2c (which is not referenced in the text), is not suited to address this question. Axonal arborization, in particular terminal arbors, are not visible in samples where multiple/all c4da axons are labeled simultaneously (Fig. S2c). Analogous to the dendrite analysis of c4da neurons single cell resolution is essential to examine this in a meaningful way. Likely, however, c4da neurons may not be a good choice to address this question.

We should be able to get single cell resolution of the c4da axon terminals using MARCM. We already have two of the knockout lines recombined with FRTs (Vps53 and Vps54) for this analysis and we will make the third recombinant line so that we can use MARCM for all three lines to examine single-cell axon morphology, as suggested.

Overall, I am concerned whether the data shown here can be generalized. The cd4a neurons are rather extreme cell types due to their very large dendritic compartment. It seems quite possible that many other neurons may not have a comparable sensitivity to the supply of lipids/sterols. This type of question can only be addressed if other types of neurons/dendrites are examined. Are class 2 or class 3 da neurons showing any defects in VPS mutants?

Given that we see the phenotype emerge during the pupal stage, we want to analyze neurons that persist from the larval to adult stages. However, not all of the dendritic arborization neurons survive into adulthood- class I and II persist, while class III die during metamorphosis (Shimono et al., 2009). As we do not have adequate tools to for studying the class II neurons, we will examine dendrite morphology of the class I neurons in larvae and adults in our knockout lines. We would be happy to look at class III neurons at the reviewers request, but our analysis will necessarily be limited to the larval stage.

Reviewer #3

Revision Plan

- Some of the experiments include multiple genotypes and so it would be important to show all in all figures. For example, figure 4B,D show four groups but figure 4F, presumably from the same set of animals, shows only three. Addition of the rescue genotype to 4F is particularly important here so should be shown. *The same concern is valid for figure 5, where puncta number and area must be available.*

The data from Fig. 4 F (using a genetically encoded marker for lysosomes, UAS-spin-RFP) are not from the same samples as Fig. 4 B and D (staining). We did not include the rescue for Fig 4. F because the lysosome marker, the rescue transgenes and the neuronal membrane marker are all on the third chromosome. We will build additional fly stocks so that we can include the rescue in experiments looking at lysosome morphology.

- This concern is amplified by the images in figure 6 of the filipin staining, that are more obviously perinuclear. However, the two sets of images in 6A and 6D, where co-staining with Golgin245 is shown, look very different. Improved images are required and it may be helpful to use supplementary information to show additional examples of the staining.

The images in Fig. 6 A are maximum projections of z stacks while Fig. 6 D shows single confocal planes, making it easier to see the perinuclear Golgi ring. Because other reviewers wanted some additional experiments related to Fig. 6 that we plan to incorporate into this figure in the revised manuscript, we will address this comment in a future revision and include additional images in the supplement.

- For the lipid regulation experiments in figure 7, please use an orthogonal approach to show that the *Osbp* and *fwd* RNAi had the expected effects on lipid accumulation.

In addition to sterol, *Osbp* and *fwd* both affect levels of PI4P at the Golgi. We have obtained a transgenic PI4P sensor that we can use to show the effect of these manipulations on this lipid as well.

3. Description of the revisions that have already been incorporated in the transferred manuscript

Reviewer #1

While the data presented clearly support a role for GARP in regulating sterol levels to

Revision Plan

support dendritic growth, they do not inter current for suffice to exclude a role for EARP as important analyses to allow such a clear cut conclusion are either insufficient or missing. If the authors wish to maintain this claim - as suggested by the title of the manuscript - further analyses are essential.

We don't mean to argue the EARP complex doesn't contribute to dendrite development at all – we do show it contributes to development in Fig 3, and as we discuss in the text.. We want to argue that the GARP and EARP complexes contribute to dendrite development by distinct mechanisms. Losing the GARP complex inhibits dendrite development by means of sterol accumulation at the TGN, which is what we are trying to highlight with our title. The reduced dendrite growth that we observe in EARP deficient neurons must occur by some other as yet unknown means. We apologize for the confusion and have reworded the title to read **“Sterol accumulates at the trans-Golgi in GARP complex deficient neurons during dendrite remodeling.”**

1- Figure 3E shows that whereas both Vps50 and Vps54 mutations reduce dendritic complexity, the Vps54 phenotype appears earlier (96h APF). Furthermore, at 7 days dendrites appear to grow again but at a slower rate than controls. This begs the question of whether these mutations are causing a delay rather than a block in the regrowth after pruning and whether the growth will eventually be normal a few days later or whether it will stop at some point.

We have included data for an additional adult timepoint (21 days) in the new Fig. 3 E. We also included graphs in which we show the statistics for each genotype over time (new Fig. S2 D-F), and discuss this analysis in the text (lines 186-195). We have also included a table of the p-values for each comparison in the Supplemental Materials (Table S2). From this analysis, we conclude that there is not a developmental delay in the knockouts, but rather a decrease in growth during the 72-96hrs APF and 1-7 day windows when the control neurons grow. We are unable to draw conclusions about the *rate* of growth as we analyzed neurons from different samples at each developmental timepoint, and not the same neurons over time.

Reviewer #2

It would be important to know, whether the dendrite morphogenesis defect is indeed a developmental patterning defect or rather a "scaling" defect due to the fact that da neurons increase their size (but not necessarily their projection pattern) during larval maturation.

Revision Plan

We have analyzed the larval data for coverage index – neuron area/hemisegment (receptive field) area as defined in (Parrish et al., 2009) to determine if there is a scaling defect at this stage in development. We do not observe a defect in scaling (Fig. S2 C) and discussed in lines 175-182.

Reviewer #3

- The statistical analyses generally look appropriate but it would be critical to clarify what N means in every case. For example in figure 2 the authors state n=8 without clarifying if this is n=8 animals or n=8 neurons. N should always be the number of animals, but then the n of independent cells counted should also be indicated. Typically, one would either pre-average per genotype or use a mixed model that includes N of animals and n of cells (or similar).

For experiments analyzing dendrite morphology, n represents the number of neurons, as we have clarified in our figure legends. As per another reviewer's request, we will increase the n for the organelle and filipin staining in our planned revision and specify fly and cell number at that time.

- Please add details of how experiments were blinded to genotype

The researcher was blinded to genotype during analysis. We have included that detail in our Methods section (line 566).

- Some of the experiments include multiple genotypes and so it would be important to show all in all figures. For example, figure 4B,D show four groups but figure 4F, presumably from the same set of animals, shows only three. Addition of the rescue genotype to 4F is particularly important here so should be shown. *The same concern is valid for figure 5, where puncta number and area must be available.*

We address the first portion of this comment in section 2, for additional experiments involving generating new fly lines. We have included data on puncta area, and mean fluorescence intensity for Rab5 and Rab7 in the supplement (Fig S3). We had already included the data on puncta number and area in Fig 5, but we have added the data on mean fluorescence intensity as well.

Revision Plan

- Related to figure 5, please provide validation of the staining of the TGN. Typically, one would expect trans Golgi to be close to the nucleus with at least some extended stacks. A Golgin245 knockout would be ideal.

The Golgi in most *Drosophila* cells is typically found as discrete puncta dispersed throughout the cytosol like what we see in the Golgin245 staining, as opposed to the ribbon "stack of pancake" morphology typically seen near the nucleus in mammalian cells. For reference, please see Figure 6D in (Ye et al., 2007), Figures 2,4,5 in (Rosa-Ferreira et al., 2015), and observations reviewed in (Kondylis and Rabouille, 2009).

The Golgin245 antibody was well characterized in the paper first describing it (Riedel et al., 2016) (colocalization with other Golgi markers, decreased staining with Golgin245 RNAi), but we would be happy to repeat this validation in the c4da neurons at the reviewer's request. There do not appear to be Golgin245 mutant or KO lines available, so we would also use the Golgin245 RNAi.

- For figures 6F, G please show examples of staining for late endosomes and lysosome with appropriate validation.

Because several of our planned revisions relate to Fig. 6, we will include images for Fig. 6 F and G when we remake this figure to incorporate those planned revisions. To clarify, we used the same reagents to mark late endosomes and lysosomes in both Fig. 4 and Fig. 6. Like the Golgin245 antibody, the Rab7 antibody was developed by the Munro lab and characterized in (Riedel et al., 2016) (partial colocalization with the endosomal marker Hrs and with the lysosomal marker Arl8). Spinster (aka benchwarmer) is a known lysosomal transmembrane protein that colocalizes with Lamp1 (Dermaut et al., 2005; Rong et al., 2011). The fluorescently tagged spin transgenes were developed by the Bellen lab and have been frequently used to mark lysosomes. We would be happy to carry out additional validation experiments at the reviewer's specification.

- The title of figure 2 is inaccurate, at least if I understand the experiment, as it does not show neuron-specific knockout but instead whole body knockout with neuron rescue. Please rephrase.

Because of the lethality of whole body $Vps53^{KO/KO}$ in adult flies, we analyze MARCM clonal neurons that are $Vps53^{KO/KO}$ in flies that are otherwise heterozygous ($Vps53^{KO/+}$). To clarify

Revision Plan

this experiment, we have changed the title of Fig. 2 from “Neuron-specific knockout of *Vps53* results in smaller dendritic arbors” to “*Vps53*^{KO/KO} MARCM clonal neurons have smaller dendritic arbors”.

- Figure 8 needs examples of the TGN and late endosome morphology.

We have included these images in Figure 8.

- For ease of reading, it would be helpful to show genotypes in the same order in all figures (see 4B, 4D)

The order appears different in Fig. 4 B & D because we only included the rescue for the KO that shows a phenotype for each staining. The genotypes included in Fig. 4 B are: +/+, *Vps50*^{KO/KO}, *Vps50*^{KO/KO} + rescue, and *Vps54*^{KO/KO}. The genotypes included in Fig. 4 D are +/+, *Vps50*^{KO/KO}, *Vps54*^{KO/KO}, *Vps54*^{KO/KO} + rescue. We have changed the shading of the bars corresponding to these rescue genotypes throughout the manuscript to make it easier to distinguish the two rescue conditions.

4. Description of analyses that authors prefer not to carry out

References Cited

- Dermaut, B., K.K. Norga, A. Kania, P. Verstreken, H. Pan, Y. Zhou, P. Callaerts, and H.J. Bellen. 2005. Aberrant lysosomal carbohydrate storage accompanies endocytic defects and neurodegeneration in *Drosophila benchwarmer*. *Journal of Cell Biology*. 170:127–139. doi:10.1083/jcb.200412001.
- Kondylis, V., and C. Rabouille. 2009. The Golgi apparatus: Lessons from *Drosophila*. *FEBS Letters*. 583:3827–3838. doi:10.1016/j.febslet.2009.09.048.

Revision Plan

- Parrish, J.Z., P. Xu, C.C. Kim, L.Y. Jan, and Y.N. Jan. 2009. The microRNA bantam Functions in Epithelial Cells to Regulate Scaling Growth of Dendrite Arbors in *Drosophila* Sensory Neurons. *Neuron*. 63:788–802. doi:10.1016/j.neuron.2009.08.006.
- Riedel, F., A.K. Gillingham, C. Rosa-Ferreira, A. Galindo, and S. Munro. 2016. An antibody toolkit for the study of membrane traffic in *Drosophila melanogaster*. *Biology Open*. 5:987–992. doi:10.1242/bio.018937.
- Rong, Y., C.K. McPhee, S. Deng, L. Huang, L. Chen, M. Liu, K. Tracy, E.H. Baehrecke, L. Yu, and M.J. Lenardo. 2011. Spinster is required for autophagic lysosome reformation and mTOR reactivation following starvation. *Proceedings of the National Academy of Sciences*. 108:7826–7831. doi:10.1073/pnas.1013800108.
- Rosa-Ferreira, C., C. Christis, I.L. Torres, and S. Munro. 2015. The small G protein Arl5 contributes to endosome-to-Golgi traffic by aiding the recruitment of the GARP complex to the Golgi. *Biology Open*. 4:474–481. doi:10.1242/bio.201410975.
- Shimono, K., A. Fujimoto, T. Tsuyama, M. Yamamoto-Kochi, M. Sato, Y. Hattori, K. Sugimura, T. Usui, K. Kimura, and T. Uemura. 2009. Multidendritic sensory neurons in the adult *Drosophila* abdomen: origins, dendritic morphology, and segment- and age-dependent programmed cell death. *Neural Dev*. 4:37. doi:10.1186/1749-8104-4-37.
- Ye, B., Y. Zhang, W. Song, S.H. Younger, L.Y. Jan, and Y.N. Jan. 2007. Growing Dendrites and Axons Differ in Their Reliance on the Secretory Pathway. *Cell*. 130:717–729. doi:10.1016/j.cell.2007.06.032.

January 11, 2022

Re: JCB manuscript #202112108T

Dr. Yuh-Nung Jan
Howard Hughes Medical Institute
Physiology
University of California, San Francisco Physiology & Biochemistry Depts. Room RH484E
1550 4th Street
San Francisco, CA 94143-0725

Dear Dr. Jan,

Thank you for submitting your manuscript entitled "Sterol accumulates at the trans-Golgi in GARP complex deficient neurons during dendrite remodeling". We have assessed your manuscript, the reviews from Review Commons, and your revision plan. We think that your work is interesting and would like to invite you to submit a revision if you can address the reviewers' key concerns, as outlined in your revision plan.

The improvements in data quality and experimental validation would be critical for a successful revision. We understand the limitations of the model, and thus we think it is not necessary to include studies in class II and III dendritic arborization neurons; analyses of dendrite morphogenesis in class I neurons in larvae and adults in the knockout lines would suffice.

GENERAL GUIDELINES:

Text limits: Character count for an Article is < 40,000, not including spaces. Count includes title page, abstract, introduction, results, discussion, acknowledgments, and figure legends. Count does not include materials and methods, references, tables, or supplemental legends.

Figures: Articles may have up to 10 main text figures. Figures must be prepared according to the policies outlined in our Instructions to Authors, under Data Presentation, <https://jcb.rupress.org/site/misc/ifora.xhtml>. All figures in accepted manuscripts will be screened prior to publication.

Supplemental information: There are strict limits on the allowable amount of supplemental data. Articles may have up to 5 supplemental figures. Up to 10 supplemental videos or flash animations are allowed. A summary of all supplemental material should appear at the end of the Materials and methods section.

Please note that JCB now requires authors to submit Source Data used to generate figures containing gels and Western blots with all revised manuscripts. This Source Data consists of fully uncropped and unprocessed images for each gel/blot displayed in the main and supplemental figures. Since your paper includes cropped gel and/or blot images, please be sure to provide one Source Data file for each figure that contains gels and/or blots along with your revised manuscript files. File names for Source Data figures should be alphanumeric without any spaces or special characters (i.e., SourceDataF#, where F# refers to the associated main figure number or SourceDataFS# for those associated with Supplementary figures). The lanes of the gels/blots should be labeled as they are in the associated figure, the place where cropping was applied should be marked (with a box), and molecular weight/size standards should be labeled wherever possible. Source Data files will be made available to reviewers during evaluation of revised manuscripts and, if your paper is eventually published in JCB, the files will be directly linked to specific figures in the published article.

As you may know, the typical timeframe for revisions is three to four months. However, we at JCB realize that the implementation of social distancing and shelter in place measures that limit spread of COVID-19 also pose challenges to scientific researchers. Lab closures especially are preventing scientists from conducting experiments to further their research.

Therefore, JCB has waived the revision time limit. We recommend that you reach out to the editors once your lab has reopened to decide on an appropriate time frame for resubmission. Please note that papers are generally considered through only one revision cycle, so any revised manuscript will likely be either accepted or rejected.

Thank you for this interesting contribution to Journal of Cell Biology. You can contact us at the journal office with any questions, cellbio@rockefeller.edu.

Sincerely,

Marc Freeman
Monitoring Editor
Journal of Cell Biology

Lucia Morgado-Palacin, PhD
Scientific Editor
Journal of Cell Biology

Revision Plan

Manuscript number: RC-2021-01087 (JCB manuscript 202112108T)

Corresponding author(s): Yuh Nung Jan

1. General Statements

We thank the reviewers for their helpful comments. We believe that we have addressed all of their concerns and suggestions. We have highlighted our responses in the revision plan and the changes we have made to the manuscript in blue text. For figures where we have added data or analyses at the request of reviewers, we have highlighted the corresponding text in the figure legends.

Reviewer #1 (Evidence, reproducibility and clarity (Required)):

Summary:

O'Brien and colleagues use *Drosophila* dendrite development to dissect the roles of the GARP and EARP vesicular trafficking complexes in the development of neuronal morphology. By making complex-specific KOs they investigate the role of each complex in the growth, pruning and re-growth of sensory dendrites and conclude that the GARP, but not EARP, complex is required for proper dendrite development by limiting sterol accumulation in the neuronal TGN.

Major comments:

While the data presented clearly support a role for GARP in regulating sterol levels to support dendritic growth, they do not inter current for suffice to exclude a role for EARP as important analyses to allow such a clear cut conclusion are either insufficient or missing. If the authors wish to maintain this claim - as suggested by the title of the manuscript - further analyses are essential.

We don't mean to argue the EARP complex doesn't contribute to dendrite development at all – we do show it contributes to development in Fig 3, and as we discuss in the text, we want to argue that the GARP and EARP complexes contribute to dendrite development by distinct mechanisms. Losing the GARP complex inhibits dendrite development by means of sterol accumulation at the TGN, which is what we are trying to highlight with the original title. The reduced dendrite growth that we observe in EARP deficient neurons must occur by some other as yet unknown means. We apologize for the confusion and have reworded the title to read "**Sterol accumulates at the trans-Golgi in GARP complex deficient neurons during dendrite remodeling.**"

1- Figure 3E shows that whereas both Vps50 and Vps54 mutations reduce dendritic

Revision Plan

complexity, the Vps54 phenotype appears earlier (96h APF). Furthermore, at 7 days dendrites appear to grow again but at a slower rate than controls. This begs the question of whether these mutations are causing a delay rather than a block in the regrowth after pruning and whether the growth will eventually be normal a few days later or whether it will stop at some point.

We have included data for an additional adult timepoint (21 days) in the new Fig. 3 E. We also included graphs in which we show the statistics for each genotype over time (new Fig. S2 D-F), and discuss this analysis in the text (lines 187-196). We have also included a table of the p-values for each comparison in the Supplemental Materials (Table S2). From this analysis, we conclude that there is not a developmental delay in the knockouts, but rather a decrease in growth during the 72-96hrs APF and 1-7 day windows when the control neurons grow. We are unable to draw conclusions about the *rate* of growth as we analyzed neurons from different samples at each developmental timepoint, and not the same neurons over time.

2- In Figure 4, the two mutations appear to have statistically differential effects on Rab5 and Rab7 puncta even though the data mean and distribution seem very similar. Interestingly, in each case the non-significant effect is associated with a smaller sample size. Given that the overall sample sizes used are rather small for such highly variable data, this could easily cause a statistical anomaly due to sampling bias. The sample size should be made uniform across all genotypes and should ideally be at least doubled.

We have increased our sample sizes for staining experiments and as per Reviewer 3's request below, clarified that the data points for these experiments represent independent samples (the average of 1-4 neurons/fly).

3- Perhaps the most important issue related to Figure 6 where the authors find that there is no sterol accumulation at 96h APF in the Vps50 mutant. However, even that the dendritic phenotype is slower to appear in this mutant compared to the Vps54, are the authors sure that the accumulation is not just slower? This should be examined using the same temporal sequence used for Vps54 shown in Figure 6C. In addition, the fact that sterol accumulation returns to normal in the Vps54 mutant at 1 day, supports the notion of a delay phenotype (see point 1 above).

These issues should be experimentally addressed to see if the data fully support the initial conclusions, or if the conclusions should be modified to suggest differential contribution of the two complexes to the process being studied and to a developmental delay phenotype.

We had included the filipin staining for *Vps50^{KO/KO}* at 1 day in Figure S4 A in the original manuscript (which did not show a significant difference from control). We have now

Revision Plan

included filipin staining data for *Vps50^{KO/KO}* in the time course in Figure 6C. As mentioned above, we do not think there is a developmental delay.

Reviewer #1 (Significance (Required)):

The study advances our understanding of the role of regulation of lipid storage in sculpting neuronal morphology during development.

Reviewer #2 (Evidence, reproducibility and clarity (Required)):

This manuscript presents a solid genetic analysis of components of the GARP and EARP complexes. The analysis is focused on a specialized type of sensory neurons i.e. class IV da neurons in *Drosophila* larvae. The authors show that loss of multiple components (*VPS50-54*) disrupt dendrite morphogenesis in c4da neurons in distinct ways. Additional genetic interaction studies further support the notion of functional differences of GARP and EARP in vivo.

Overall this is a solid study and with one exception (see below) I have little concern regarding the presented experiments. I do, however, find the exclusive focus on a highly specialized cell type c4da somewhat problematic.

Concerns:

Experimental concern: It is stated that loss of *VPS50* and *VPS54* only causes dendrite morphogenesis defects. However, the corresponding supplemental figure S2c (which is not referenced in the text), is not suited to address this question. Axonal arborization, in particular terminal arbors, are not visible in samples where multiple/all c4da axons are labeled simultaneously (Fig. S2c). Analogous to the dendrite analysis of c4da neurons single cell resolution is essential to examine this in a meaningful way. Likely, however, c4da neurons may not be a good choice to address this question.

We recombined the *Vps53^{KO}* and *Vps54^{KO}* lines with *FRT^{40A}* and crossed them to the yw SOP-FLP; *FRT^{40A}* tubGal80; ppk-Gal4, UAS-CD4-tdGFP stock to generate c4da MARCM clones in which to examine the morphology of individual axon terminals. Please see the Methods section (lines 583-596) and Figure S2 G and H.

We recombined *Vps50^{KO}* with *FRT^{42D}* and attempted but ultimately failed to establish the same MARCM stock as above, exchanging *FRT^{40A}* with *FRT^{42D}*. The other MARCM stock available to use with *FRT^{42D}* was yw SOP-FLP; *FRT^{42D}* tubGal80; nsyb-Gal4, UAS-tdTomato. As

Revision Plan

the nsyb-Gal4 was able to generate a much larger number of clones than the c4da-specific ppk-Gal4, we were not confident that we could isolate the c4da axon terminals for analysis.

As Vps54 is specific to the GARP complex, and Vps53 is found in both the GARP and EARP complexes, we believe these experiments allow us to conclude that neither complex is required for axon development. We are unable to draw any conclusions about any role Vps50 by itself may play, independently of Vps53, in axons, as we discuss in lines 197-204 and lines 346-351.

It would be important to know, whether the dendrite morphogenesis defect is indeed a developmental patterning defect or rather a "scaling" defect due to the fact that da neurons increase their size (but not necessarily their projection pattern) during larval maturation.

We have analyzed the larval data for coverage index – neuron area/hemisegment (receptive field) area as defined in (Parrish et al., 2009) to determine if there is a scaling defect at this stage in development. We do not observe a defect in scaling (Fig. S2 C) and discussed in lines 176-179).

Overall, I am concerned whether the data shown here can be generalized. The cd4a neurons are rather extreme cell types due to their very large dendritic compartment. It seems quite possible that many other neurons may not have a comparable sensitivity to the supply of lipids/sterols. This type of question can only be addressed if other types of neurons/dendrites are examined. Are class 2 or class 3 da neurons showing any defects in VPS mutants?

Given that we see the phenotype emerge during the pupal stage, we wanted to analyze neurons that persist at least into the adulthood. However, not all of the dendritic arborization neurons survive this long - class I and II persist at least temporarily, while class III die during metamorphosis (Shimono et al., 2009). As we do not have adequate tools to for studying the class II neurons, we examined class I da neuron morphology in larvae and adults. Please see Figure 4 and discussion in lines 205-211 and lines 334- 345. Briefly, we found that after pruning and regrowth, *Vps54^{KO/KO}* but not *Vps50^{KO/KO}* c1da neurons in adults had reduced dendrite branch length, suggesting the requirement of GARP complex for dendrite development is not limited to c4da neuron.

Reviewer #2 (Significance (Required)):

At this point i am not convinced that the findings can be generalized. The c4da neuron is really an extreme cell type with a massive disproportionate increase in membrane

Revision Plan

extensions. This is rather unusual and other neuron types should be tested.

Reviewer #3 (Evidence, reproducibility and clarity (Required)):

O'Brien et al report how deficiency in GARP specific protein VPS54 or the EARP specific protein VPS50 affects the developmental dendritic remodeling of multidendritic class IV da (c4da) neurons in *Drosophila*. The main findings are that while both complexes play a role in dendritic remodeling, VPS54 deficiency leads to lipid accumulation in the trans-Golgi network (TGN). Manipulating sterols at the TGN affects dendritic remodeling suggesting that lipid accumulation is responsible for control of neuron morphology in this model. Overall, the data is interesting and the authors develop the experiments enough to be convincing on their major claims. However, a few aspects need clarification and perhaps revisiting conclusions.

Major comments

- The statistical analyses generally look appropriate but it would be critical to clarify what N means in every case. For example in figure 2 the authors state n=8 without clarifying if this is n=8 animals or n=8 neurons. N should always be the number of animals, but then the n of independent cells counted should also be indicated. Typically, one would either pre-average per genotype or use a mixed model that includes N of animals and n of cells (or similar).

For morphological analysis of neurons, we traced individual neurons in separate larvae or flies, and therefore the n represents independent neurons as samples. For organelle and filipin staining, each data point represents 1-4 neurons/fly (averaged). We have updated our figure legends and methods section (lines 580-581 and 613-614) to reflect this. As per reviewer 1's request, we also increased our sample size for staining experiments.

- Please add details of how experiments were blinded to genotype

The researcher was blinded to genotype during data analysis. We have included that detail in our Methods section (line 617-618).

- Some of the experiments include multiple genotypes and so it would be important to show all in all figures. For example, figure 4B,D show four groups but figure 4F, presumably from the same set of animals, shows only three. Addition of the rescue genotype to 4F is particularly important here so should be shown. The same concern is valid for figure 5, where puncta number and area must be available.

The data from the original Fig. 4 F (using a genetically encoded marker for lysosomes, UAS-Spin-RFP) are not from the same samples as Fig. 4 B and D (staining). We did not include

Revision Plan

the rescue for Fig 4. F because the lysosome marker Spin-RFP, the rescue transgenes and the neuronal membrane marker are all on the third chromosome. In order to include a rescue for the lysosome phenotype in *Vps54*^{KO/KO} neurons, we recombined *Vps50*^{KO} and *Vps54*^{KO} with UAS-GFP-Lamp, another well characterized and widely used lysosome marker. This allowed us to establish crosses with the recombinants, rescue transgenes and neuronal membrane markers in the same experiment, the results of which we now report in the new Fig. 5E and F. We have moved the Spin-RFP (without rescue) to the supplement (Fig S3 M-O). In Fig 5, we report puncta number for each organelle, and include the puncta area and mean fluorescence intensity for each in the Fig S3.

- Related to figure 5, please provide validation of the staining of the TGN. Typically, one would expect trans Golgi to be close to the nucleus with at least some extended stacks. A Golgin245 knockout would be ideal.

The Golgi in most *Drosophila* cells is typically found as discrete puncta dispersed throughout the cytosol like what we see in the Golgin245 staining, as opposed to the ribbon "stack of pancake" morphology typically seen near the nucleus in mammalian cells. For reference, please see Figure 6D in (Ye et al., 2007), Figures 2,4,5 in (Rosa-Ferreira et al., 2015), and observations reviewed in (Kondylis and Rabouille, 2009). The Golgin245 antibody was well characterized in the paper first describing it (Riedel et al., 2016) (colocalization with other Golgi markers, decreased staining with Golgin245 RNAi).

- This concern is amplified by the images in figure 6 of the filipin staining, that are more obviously perinuclear. However, the two sets of images in 6A and 6D, where co-staining with Golgin245 is shown, look very different. Improved images are required and it may be helpful to use supplementary information to show additional examples of the staining.

The images in the original Fig. 6 A are maximum projections of z stacks while Fig. 6 D shows single confocal planes, making it easier to see the perinuclear Golgi ring. We have changed this figure to now show the maximum intensity projections in Fig. 6A (as before, but with better images) to show the overall filipin levels. To show filipin staining in organelles, we have now included the maximum intensity projections of the total filipin staining, organelle marker (used to create the mask to extract the organelle-specific filipin signal), and the maximum intensity projection of the extracted organelle-specific filipin signal in Fig. 6 D and F, and Fig. S4 A and C.

- For figures 6F, G please show examples of staining for late endosomes and lysosome with appropriate validation.

We used the same reagents to mark late endosomes and lysosomes in both Fig. 4, Fig S3, and Fig. 6 (now Fig. 7). Like the Golgin245 antibody, the Rab7 antibody was developed by

Revision Plan

the Munro lab and characterized in (Riedel et al., 2016) (partial colocalization with the endosomal marker Hrs and with the lysosomal marker Arl8). Spinster (aka benchwarmer) is a known lysosomal transmembrane protein that colocalizes with Lamp1 (Dermaut et al., 2005; Rong et al., 2011). The fluorescently tagged spin transgenes were developed by the Bellen lab and have been frequently used to mark lysosomes. We have also included lysosome morphology data with the more ubiquitous lysosome marker GFP-Lamp (Fig. 4 E and F).

As mentioned in the above point, we have now included images of total filipin staining, organelle staining and the extracted organelle-specific filipin staining in Fig. 6 D and F, and Fig. S4 A and C.

- For the lipid regulation experiments in figure 7, please use an orthogonal approach to show that the *Osbp* and *fwd* RNAi had the expected effects on lipid accumulation.

In addition to regulating sterol levels, *Osbp* and *fwd* PI4P affect levels as well. While we had already included filipin staining of sterol levels in the original manuscript (Fig 7), we have now also included data using the PI4P sensor P4M-GFP (Balakrishnan et al., 2018), Fig. 7 I-K and discussed the results in lines 309-314.

- Figure 8 needs examples of the TGN and late endosome morphology.

We have included these images in Figure 8.

Minor comments

- The title of figure 2 is inaccurate, at least if I understand the experiment, as it does not show neuron-specific knockout but instead whole body knockout with neuron rescue. Please rephrase.

Because of the lethality of whole body *Vps53^{KO/KO}* in adult flies, we analyzed MARCM clonal neurons that are *Vps53^{KO/KO}* in flies that are otherwise heterozygous (*Vps53^{KO/+}*). Because the MARCM technique only drives transgenes in the homozygous KO clonal neuron, the rescue transgene is therefore only expressed in the clones as well. To clarify this experiment, we have changed the title of Fig. 2 from "Neuron-specific knockout of *Vps53* results in smaller dendritic arbors" to "*Vps53^{KO/KO}* MARCM clonal neurons have smaller dendritic arbors".

- For ease of reading, it would be helpful to show genotypes in the same order in all figures (see 4B, 4D)

The order appears different in Fig. 4 B & D because we only included the rescue for the KO that shows a phenotype for each staining. The genotypes included in the Fig. 4 (now the new Fig. 5) B (Rab5 staining) are: +/+, *Vps50^{KO/KO}*, *Vps50^{KO/KO}* + rescue, and *Vps54^{KO/KO}*. The

Revision Plan

genotypes included in Fig. 4 D (Rab7 staining), F (GFP-Lamp) and H (Golgin145 staining) are +/+, *Vps50*^{KO/KO}, *Vps54*^{KO/KO}, *Vps54*^{KO/KO} + rescue. We have changed the color scheme of the rescue genotypes throughout the manuscript to make it easier to distinguish the rescue conditions.

Reviewer #3 (Significance (Required)):

The advance here is to nominate lipid accumulation at the trans Golgi network (TGN) is sufficient to affect dendritic remodeling during development. Although the work is performed in a model system, it may have relevance to human neurodevelopmental disorders caused by mutations in the orthologous genes. The work will be of highest relevance to developmental neurobiologists, particularly those working on GARP or EARP mutations and those who use *Drosophila* as an appropriate model for neurodevelopment.

Balakrishnan, S.S., U. Basu, D. Shinde, R. Thakur, M. Jaiswal, and P. Raghu. 2018. Regulation of PI4P levels by PI4KIII α during G-protein coupled PLC signaling in *Drosophila* photoreceptors. *Journal of Cell Science*. jcs.217257. doi:10.1242/jcs.217257.

Dermaut, B., K.K. Norga, A. Kania, P. Verstreken, H. Pan, Y. Zhou, P. Callaerts, and H.J. Bellen. 2005. Aberrant lysosomal carbohydrate storage accompanies endocytic defects and neurodegeneration in *Drosophila benchwarmer*. *Journal of Cell Biology*. 170:127–139. doi:10.1083/jcb.200412001.

Kondylis, V., and C. Rabouille. 2009. The Golgi apparatus: Lessons from *Drosophila*. *FEBS Letters*. 583:3827–3838. doi:10.1016/j.febslet.2009.09.048.

Parrish, J.Z., P. Xu, C.C. Kim, L.Y. Jan, and Y.N. Jan. 2009. The microRNA bantam Functions in Epithelial Cells to Regulate Scaling Growth of Dendrite Arbors in *Drosophila* Sensory Neurons. *Neuron*. 63:788–802. doi:10.1016/j.neuron.2009.08.006.

Riedel, F., A.K. Gillingham, C. Rosa-Ferreira, A. Galindo, and S. Munro. 2016. An antibody toolkit for the study of membrane traffic in *Drosophila melanogaster*. *Biology Open*. 5:987–992. doi:10.1242/bio.018937.

Rong, Y., C.K. McPhee, S. Deng, L. Huang, L. Chen, M. Liu, K. Tracy, E.H. Baehrecke, L. Yu, and M.J. Lenardo. 2011. Spinster is required for autophagic lysosome reformation and mTOR reactivation following starvation. *Proceedings of the National Academy of Sciences*. 108:7826–7831. doi:10.1073/pnas.1013800108.

Revision Plan

Rosa-Ferreira, C., C. Christis, I.L. Torres, and S. Munro. 2015. The small G protein Arl5 contributes to endosome-to-Golgi traffic by aiding the recruitment of the GARP complex to the Golgi. *Biology Open*. 4:474–481. doi:10.1242/bio.201410975.

Shimono, K., A. Fujimoto, T. Tsuyama, M. Yamamoto-Kochi, M. Sato, Y. Hattori, K. Sugimura, T. Usui, K. Kimura, and T. Uemura. 2009. Multidendritic sensory neurons in the adult *Drosophila* abdomen: origins, dendritic morphology, and segment- and age-dependent programmed cell death. *Neural Dev*. 4:37. doi:10.1186/1749-8104-4-37.

Ye, B., Y. Zhang, W. Song, S.H. Younger, L.Y. Jan, and Y.N. Jan. 2007. Growing Dendrites and Axons Differ in Their Reliance on the Secretory Pathway. *Cell*. 130:717–729. doi:10.1016/j.cell.2007.06.032.

September 14, 2022

RE: JCB Manuscript #202112108R

Dr. Yuh-Nung Jan
Howard Hughes Medical Institute
Physiology
University of California, San Francisco Physiology & Biochemistry Depts. Room RH484E
1550 4th Street
San Francisco, CA 94143-0725

Dear Dr. Jan:

Thank you for submitting your revised manuscript entitled "Sterol accumulates at the trans-Golgi in GARP complex deficient neurons during dendrite remodeling". The reviewers have now assessed your revised manuscript and, as you can see, they are satisfied with revisions. Thus, we would be happy to publish your paper in JCB pending final revisions necessary to meet our formatting guidelines (see details below).

To avoid unnecessary delays in the acceptance and publication of your paper, please read the following information carefully. Please go through all the formatting points paying special attention to those marked with asterisks.

A. MANUSCRIPT ORGANIZATION AND FORMATTING:

1) Text limits: Character count for Articles and Tools is < 40,000, not including spaces. Count includes title page, abstract, introduction, results, discussion, and acknowledgments. Count does not include materials and methods, figure legends, references, tables, or supplemental legends.

2) Figures limits: Articles and Tools may have up to 10 main text figures.

***** Please note that main text figures should be provided as individual, editable files.**

3) Figure formatting:

***** Molecular weight or nucleic acid size markers must be included on all gel electrophoresis. Please, include size markers in main Fig. 1B and supplemental Figs. 1C, 1E, 1G, 1H, 3R (tubulin blot).**

Scale bars must be present on all microscopy images, including inset magnifications.

Also, please avoid pairing red and green for images and graphs to ensure legibility for color-blind readers. If red and green are paired for images, please ensure that the particular red and green hues used in micrographs are distinctive with any of the colorblind types. If not, please modify colors accordingly or provide separate images of the individual channels.

4) Statistical analysis:

Error bars on graphic representations of numerical data must be clearly described in the figure legend.

The number of independent data points (n) represented in a graph must be indicated in the legend. Please, also indicate whether 'n' refers to technical or biological replicates (i.e. number of analyzed cells, samples or animals, number of independent experiments).

Statistical methods should be explained in full in the materials and methods in a separate section.

For figures presenting pooled data the statistical measure should be defined in the figure legends.

Please also be sure to indicate the statistical tests used in each of your experiments (both in the figure legend itself and in a

separate methods section) as well as the parameters of the test (for example, if you ran a t-test, please indicate if it was one- or two-sided, etc.).

*** As you used parametric tests in your study (i.e. t-tests), you should have first determined whether the data was normally distributed before selecting that test. In the stats section of the methods, please indicate how you tested for normality. If you did not test for normality, you must state something to the effect that "Data distribution was assumed to be normal but this was not formally tested."

5) Abstract and title:

The abstract should be no longer than 160 words and should communicate the significance of the paper for a general audience.

*** The title should be less than 100 characters including spaces. Make the title concise but accessible to a general readership. We would like to make the following suggestion for the title: "The GARP complex prevents sterol accumulation at the trans-Golgi during dendrite remodeling".

6) Materials and methods:

Should be comprehensive and not simply reference a previous publication for details on how an experiment was performed. The text should not refer to methods "...as previously described."

Also, the materials and methods should be included with the main manuscript text and not in the supplementary materials.

7) Please be sure to provide the sequences for all of your primers/oligos and RNAi constructs in the materials and methods.

*** You must also indicate in the methods the source, species, and catalog numbers (where appropriate) for all of your antibodies. Please include the species for all your antibodies.

8) Microscope image acquisition:

The following information must be provided about the acquisition and processing of images:

a. Make and model of microscope

b. Type, magnification, and numerical aperture of the objective lenses

c. Temperature

d. imaging medium

e. Fluorochromes

*** f. Camera make and model

g. Acquisition software

h. Any software used for image processing subsequent to data acquisition. Please include details and types of operations involved (e.g., type of deconvolution, 3D reconstitutions, surface or volume rendering, gamma adjustments, etc.).

10) Supplemental materials:

There are strict limits on the allowable amount of supplemental data. Articles/Tools may have up to 5 supplemental figures. There is no limit for supplemental tables.

Please note that supplemental figures and tables should be provided as individual, editable files.

*** A summary of all supplemental material should appear at the end of the Materials and Methods section (please see any recent JCB paper for an example of this summary).

11) eTOC summary:

A ~40-50 word summary that describes the context and significance of the findings for a general readership should be included on the title page.

The statement should be written in the present tense and refer to the work in the third person. It should begin with "First author name(s) et al..." to match our preferred style.

12) Conflict of interest statement:

JCB requires inclusion of a statement in the acknowledgements regarding competing financial interests. If no competing financial interests exist, please include the following statement: "The authors declare no competing financial interests."

13) A separate author contribution section is required following the Acknowledgments in all research manuscripts.

*** All authors should be mentioned and designated by their first and middle initials and full surnames and the CRediT nomenclature is encouraged (<https://casrai.org/credit/>).

14) ORCID IDs: ORCID IDs are unique identifiers allowing researchers to create a record of their various scholarly contributions in a single place. At resubmission of your final files, please consider providing an ORCID ID for as many contributing authors as possible.

15) Materials and data sharing:

All animal and human studies must be conducted in compliance with relevant local guidelines, such as the US Department of Health and Human Services Guide for the Care and Use of Laboratory Animals or MRC guidelines, and must be approved by the authors' Institutional Review Board(s). A statement to this effect with the name of the approving IRB(s) must be included in the Materials and Methods section.

*** As a condition of publication, authors must make protocols and unique materials (including, but not limited to, cloned DNAs; antibodies; bacterial, animal, or plant cells; and viruses) described in our published articles freely available upon request by researchers, who may use them in their own laboratory only. All materials must be made available on request and without undue delay. Please, indicate whether the fly strains and reagents generated in this study have been deposited in public repositories. If not, please state that they would be made available to the scientific community upon request in the 'Data availability' section.

All datasets included in the manuscript must be available from the date of online publication, and the source code for all custom computational methods, apart from commercial software programs, must be made available either in a publicly available database or as supplemental materials hosted on the journal website. Numerous resources exist for data storage and sharing (see Data Deposition: <https://rupress.org/jcb/pages/data-deposition>), and you should choose the most appropriate venue based on your data type and/or community standard. If no appropriate specific database exists, please deposit your data to an appropriate publicly available database.

16) Please note that JCB now requires authors to submit Source Data used to generate figures containing gels and Western blots with all revised manuscripts. This Source Data consists of fully uncropped and unprocessed images for each gel/blot displayed in the main and supplemental figures. The Source Data files will be directly linked to specific figures in the published article.

Since your paper includes cropped gel and/or blot images, please be sure to provide one Source Data file for each figure that contains gels and/or blots along with your revised manuscript files. File names for Source Data figures should be alphanumeric without any spaces or special characters (i.e., SourceDataF#, where F# refers to the associated main figure number or SourceDataFS# for those associated with Supplementary figures). The lanes of the gels/blots should be labeled as they are in the associated figure, the place where cropping was applied should be marked (with a box), and molecular weight/size standards should be labeled wherever possible.

B. FINAL FILES:

Thank you for your attention to these final processing requirements. Please revise and format the manuscript and upload materials within 7 days. Please let us know if any complication preventing you from meeting this deadline arises and we can work with you to determine a suitable revision period.

Please contact the journal office with any questions, cellbio@rockefeller.edu.

Thank you for this interesting contribution, we look forward to publishing your paper in Journal of Cell Biology.

Sincerely,

Marc Freeman
Monitoring Editor
Journal of Cell Biology

Lucia Morgado-Palacin, PhD
Scientific Editor
Journal of Cell Biology

Reviewer #1 (Comments to the Authors (Required)):

The authors have addressed my concerns. I have no further comments.

Reviewer #3 (Comments to the Authors (Required)):

I believe the authors have adequately responded to my prior concerns.